# SLX4IP and telomere dynamics dictate breast cancer metastasis and therapeutic responsiveness

Nathaniel J Robinson[1], Chevaun D Morrison-Smith[2], Alex J Gooding[1], Barbara J Schiemann[2], Mark W Jackson[1], Derek J Taylor[3], William P Schiemann[2]

Metastasis is the leading cause of breast cancer-related death and poses a substantial clinical burden owing to a paucity of targeted treatment options. The clinical manifestations of metastasis occur years-to-decades after initial diagnosis and treatment because disseminated tumor cells readily evade detection and resist therapy, ultimately giving rise to recurrent disease. Using an unbiased genetic screen, we identified SLX4-interacting protein (SLX4IP) as a regulator of metastatic recurrence and established its relationship in governing telomere maintenance mechanisms (TMMs). Inactivation of SLX4IP suppressed alternative lengthening of telomeres (ALT), coinciding with activation of telomerase. Importantly, TMM selection dramatically influenced metastatic progression and survival of patients with genetically distinct breast cancer subtypes. Notably, pharmacologic and genetic modulation of TMMs elicited telomere-dependent cell death and prevented disease recurrence by disseminated tumor cells. This study illuminates SLX4IP as a potential predictive biomarker for breast cancer progression and metastatic relapse. SLX4IP expression correlates with TMM identity, which also carries prognostic value and informs treatment selection, thereby revealing new inroads into combating metastatic breast cancers.

## Introduction

Globally, breast cancer is the most commonly diagnosed malignancy and the most common cause of cancer-related death in women (Bray et al, 2018). The challenges imposed by this tremendous clinical burden are amplified by metastasis, which occurs in up to 30 percent of breast cancer cases (Cianfrocca & Goldstein, 2004). Metastasis is a multistep cascade commencing with migration from the primary tumor site and terminating in seeding and colonization of distant organs. Despite significant advances in diagnosis and treatment, metastasis remains the cause of ~90 percent of breast cancer mortality (Chaffer & Weinberg, 2011). Metastatic breast cancer cells

possess insidious properties that facilitate their escape from the primary site at early stages of tumor formation and promote their perpetuation and outgrowth upon arrival at metastatic niches. Emerging evidence indicates that disseminated breast cancer cells respond to cell-intrinsic, microenvironmental, and systemic cues to enable their prolonged survival and eventual expansion, culminating in disease recurrence and untoward patient outcomes (Nguyen & Massague, 2007; Redig & McAllister, 2013). Nevertheless, the complex molecular mechanisms that underlie metastasis remain incompletely understood, thus limiting the design and implementation of targeted therapeutic strategies.

Enabling replicative immortality is a critical step in malignant transformation and disease progression. This is primarily achieved via extension of telomeres (Hanahan & Weinberg, 2011). In many cancers, telomeres are extended by telomerase, a ribonucleo-protein composed of a reverse transcriptase and an RNA template. A growing body of evidence suggests that telomerase activation preferentially influences the metastatic potential of cancer cells (Robinson & Schiemann, 2016), and that nonproliferative disseminated tumor cells (DTCs) exhibit decreased telomerase activity (Pfitzenmaier et al, 2006). In contrast, a subset of cancers relies upon alternative lengthening of telomeres (ALT) for telomere extension (Heaphy et al, 2011b). ALT requires transient deprotection of telomeres to activate a DNA damage response (DDR) that facilitates homology-directed, recombination-dependent DNA replication (Kamranvar et al, 2013; Dilley et al, 2016). At present, the relationship between ALT and metastasis is not well characterized, and as such, elucidating the molecular functions of telomere maintenance mechanisms (TMMs) in metastasis will provide critical pathophysiologic insight.

In this study, we used validation-based insertional mutagenesis (VBIM) (Lu et al, 2009) to identify genetic regulators of breast cancer metastatic outgrowth and disease recurrence. In doing so, we discovered that SLX4-interacting protein (SLX4IP) controls the propensity of DTCs to initiate metastatic outgrowth. Moreover, SLX4IP expression patterns are associated with specific TMMs, which readily influence the metastatic properties of breast cancer cells and their sensitivity to specific telomere-targeting agents. Collectively, these

[1]Department of Pathology, Case Western Reserve University School of Medicine, Cleveland, OH, USA  [2]Case Comprehensive Cancer Center, Case Western Reserve University, Cleveland, OH, USA  [3]Department of Pharmacology, Case Western Reserve University School of Medicine, Cleveland, OH, USA

Correspondence: wps20@case.edu

findings have identified new inroads to potentially alleviate metastatic breast cancers.

## Results

### SLX4IP regulates the outgrowth properties of metastatic breast cancer cells

To identify genes that initiate metastatic recurrence, we performed VBIM using a dual in vitro–in vivo screening approach in dormant murine D2.OR breast cancer cells (Fig S1A; [Morris et al, 1994]). VBIM lentiviruses contain a strong (CMV) mutagenic promoter and a fluorescent reporter (GFP). Upon integration, the proviral DNA is flanked by LoxP sites, which allows for Cre recombinase–mediated excision of the promoter to distinguish insertional mutants (so-called "convertants") from spontaneous mutants (Lu et al, 2009). We screened D2.OR ($6 \times 10^6$) cells with an expected convertant frequency of 0.001%. This procedure yielded 48 putative metastatic clones that were initially selected from three-dimensional (3D) culture based on morphological characteristics, GFP fluorescence, and organoid outgrowth (Fig S1B). Of these, three clones were injected intravenously into BALB/c mice and monitored for pulmonary tumor formation. One clone (VBIM 2-1) exhibited robust metastatic outgrowth compared with parental D2.OR cells (Fig S1C). Importantly, the observed behavior of the VBIM 2-1 clone was reliant upon VBIM, as evidenced by reinstatement of the parental phenotype upon removal of the VBIM construct (Fig S1D and E).

Unbiased amplification of VBIM-associated transcripts in this clone revealed *SLX4-interacting protein* (*SLX4IP*, also known as *c20orf94* in humans) as the target of insertional mutagenesis. Insertion was mapped to exon 12 (*Mus musculus* chr2:137,067,593-137,068,031) of the *SLX4IP* open reading frame. SLX4IP showed a 50% reduction in expression in these cells, consistent with heterozygous loss of function (Fig S1F and G). The RNA product of VBIM-driven transcription was a synthetic antisense transcript (asSLX4IP) that was highly unstable (Fig S1H), suggesting that it lacks regulatory capacity. As a means of independent validation, we generated D2.OR derivatives in which SLX4IP expression was reduced ~50% by RNA interference (shSLX4IP-1; shSLX4IP-2) and subjected them to the same in vitro and in vivo assays used in our VBIM screen (Fig 1A–D). In line with our initial findings, SLX4IP-depleted D2.OR cells showed enhanced 3D-outgrowth (Fig 1B and C) and pulmonary tumor formation (Fig 1D) as compared with their parental counterparts. Importantly, reconstituting SLX4IP expression in SLX4IP-depleted D2.OR cells produced outgrowth dynamics that mirrored parental cells (Fig 1E and F). Coinciding with these changes in D2.OR cell behavior were dramatic transcriptomic alterations stemming from SLX4IP knockdown. Indeed, microarray analyses revealed that cells deficient in SLX4IP showed a marked reduction in the expression of genes that inhibit metastasis (Fig S2). Taken together, these findings demonstrate that SLX4IP negatively regulates the metastatic outgrowth of D2.OR cells; they also show that inactivation of SLX4IP may provide dormant DTCs with a means to reactivate proliferative programs and recur.

### SLX4IP expression patterns are associated with distinct TMMs

SLX4IP is a member of the SLX4 structure-specific endonuclease (SSE) complex (Svendsen et al, 2009). Recently, SLX4IP was shown to modulate the activity of SLX4-associated proteins involved in interstrand DNA crosslink repair (Zhang et al, 2019) and homologous recombination (Panier et al, 2019). Although microdeletions in the SLX4IP gene have been associated with acute lymphoblastic leukemia (Meissner et al, 2014), the molecular functions of SLX4IP in the context of cancer remain poorly understood. SLX4 and its associated nucleases have been implicated in the regulation of telomere length and stability (Wan et al, 2013). Accordingly, we observed SLX4IP to localize to telomeres (Fig 2A). Along these lines, several factors responsible for telomere maintenance were enriched amongst the cohort of differentially expressed genes after SLX4IP knockdown (Fig S3). Of particular interest, we found that depletion of SLX4IP produced a concomitant increase in the expression of the telomerase reverse transcriptase subunit (TERT; Fig 2B), as well as the formation (Fig 2C) and activity (Fig 2D) of the core telomerase holoenzyme (i.e., TERT plus the telomerase RNA component, TR). Importantly, loss of SLX4IP was both necessary and sufficient for telomerase activation, an event that could be reversed after restoration of SLX4IP expression in SLX4IP-deficient D2.OR cells (Fig 2E). Similar up-regulation of TERT was elicited when SLX4IP was suppressed in U2OS cells (Fig 2F and G), which are ordinarily not reliant upon telomerase (Bryan et al, 1997). Finally, loss of SLX4IP initially resulted in transient telomere shortening consistent with TMM loss, followed by substantial telomere extension concurrent with telomerase activation (Fig 2H). Although the calculated telomere lengths are shorter than would normally be expected for murine cell lines, many of these telomere length calculations derive from studies conducted on primary mouse cells or tissues (Zijlmans et al, 1997). In contrast, several studies found that established murine cell lines possess telomeres that are similar in length to those in human cells, and to those measured in our D2.OR derivatives (McIlrath et al, 2001; Sachsinger et al, 2001; Marie-Egyptienne et al, 2008).

The finding that parental D2.OR cells possess low telomerase activity appears at odds with the notion that cancer cells require telomere extension to achieve replicative immortality (Hanahan & Weinberg, 2011). Although telomerase is believed to be the dominant TMM in many cancers, telomeres may also be extended via the homologous recombination-based ALT pathway (Bryan et al, 1997). Indeed, previous examinations of patient-derived tumor specimens revealed a subset of breast cancer cases in which ALT was detected (Subhawong et al, 2009; Heaphy et al, 2011b). ALT exhibits several cardinal features that can be assessed experimentally: (i) localization of the promyelocytic leukemia (PML) protein to telomeres (ALT-associated PML bodies or APBs; [Yeager et al, 1999]), (ii) presence of extrachromosomal circular DNAs containing telomeric repeat sequences (C-circles; [Henson et al, 2009]), and (iii) loss of expression of the chromatin remodelers, ATRX and DAXX (Heaphy et al, 2011a). With these in mind, we assessed whether D2.OR cells are normally reliant upon ALT to maintain their telomeres. Interestingly, parental D2.OR cells exhibited multiple hallmarks of ALT[+] cells, including abundant APB (Fig 3A and B) and C-circle (Fig 3C) formation. In addition, ATRX and DAXX were transcriptionally silenced in these cells (Fig 3D and E).

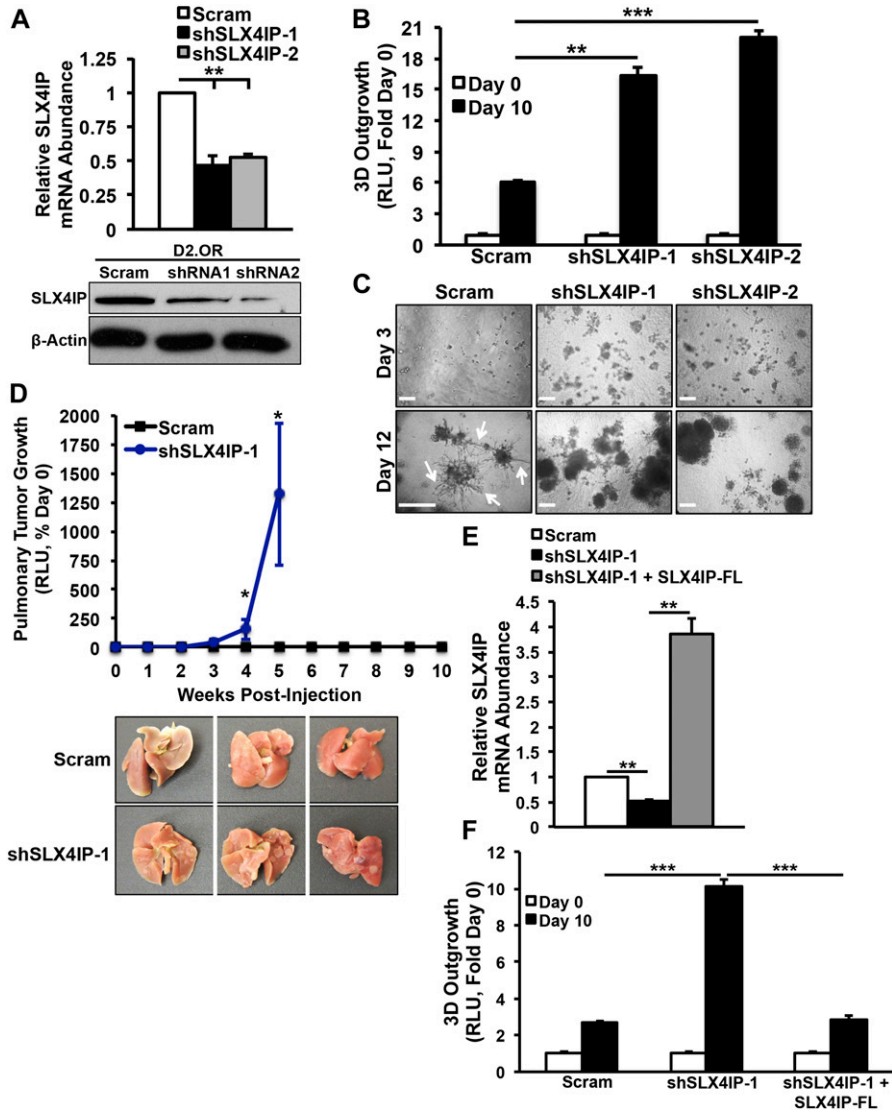

**Figure 1. SLX4IP regulates the outgrowth properties of metastatic breast cancer cells.**
**(A)** Validation of SLX4IP knockdown in D2.OR cells by qRT-PCR (*top*) and immunoblot (*bottom*). In the Western blot image, shSLX4IP-1 and shSLX4IP-2 are notated as shRNA1 and shRNA2, respectively. *n* = 3 for qRT-PCR. **(B)** Quantitation of 3D-outgrowth of parental and SLX4IP-depleted D2.OR cells (*n* = 4). **(C)** Representative images of organoids formed by indicated cell lines in 3D-culture. Arrows highlight stellate morphology characteristic of dormant D2.OR cells. Scale bar: 25 μm. **(D)** *Top*: bioluminescence imaging of pulmonary tumor formation in mice inoculated with designated D2.OR derivatives. *Bottom*: lungs harvested from mice inoculated with indicated D2.OR derivatives (*n* = 5). **(E)** qRT-PCR of SLX4IP mRNA in SLX4IP-reconstituted D2.OR cells, confirming successful ectopic expression. FL: full-length. *n* = 3. **(F)** Quantitation of 3D outgrowth of parental, SLX4IP-depleted, and SLX4IP-reconstituted D2.OR cells (*n* = 4). **(A, B, D, E, F)** *$P < 0.05$, **$P < 0.01$, ***$P < 0.001$, Mann–Whitney *U* test (Panel D) or Kruskal–Wallis test (Panels A, B, E, and F).

In stark contrast, SLX4IP knockdown resulted in loss of C-circles and APBs and activation of ATRX and DAXX expression (Fig 3A–E). Remarkably, APB formation was restored in SLX4IP-deficient D2.OR cells after reconstitution of SLX4IP expression (Fig 3A). Similar effects were elicited when SLX4IP was inactivated in ALT[+] U2OS cells (Fig 3F and G). Collectively, these findings denote an association between SLX4IP and telomere homeostasis and, together with our investigations connecting SLX4IP to metastasis (Fig 1), suggest a broader relationship between TMM selection and metastatic recurrence.

### Telomerase is required for the acquisition of metastatic features after SLX4IP inactivation

Because both ALT and telomerase ostensibly achieve the same end (i.e., telomere lengthening), we examined whether the up-regulated expression and activity of telomerase was necessary for metastatic outgrowth in cells rendered deficient in SLX4IP expression. Using CRISPR/Cas9, we deleted TERT in SLX4IP-deficient D2.OR cells to ablate their induction of TERT elicited by SLX4IP loss (Fig 4A). Importantly, these derivatives were substantially impaired in their ability to grow both in 3D-culture (Fig 4B) and in vivo (Fig 4C), indicating that telomerase is essential in driving metastatic recurrence in cells rendered deficient in SLX4IP.

In the absence of both SLX4IP and TERT, cells may have no recourse for telomere maintenance, and consequently, these TMM-deficient cells would be expected to undergo telomere attrition culminating in p21-dependent, p16-independent senescence (Herbig et al, 2004). Accordingly, we observed significant (i) telomere shortening (Fig 4D), (ii) up-regulation of p21 expression (Fig 4E), and (iii) enhancement of senescence-associated (SA) β-galactosidase expression and activity (Fig 4F and G) in TMM-deficient cells. Collectively, these findings suggest a potential direct role of SLX4IP in promoting ALT and reinforce the notion that activation of TMMs in DTCs plays an important and multifaceted role in metastasis.

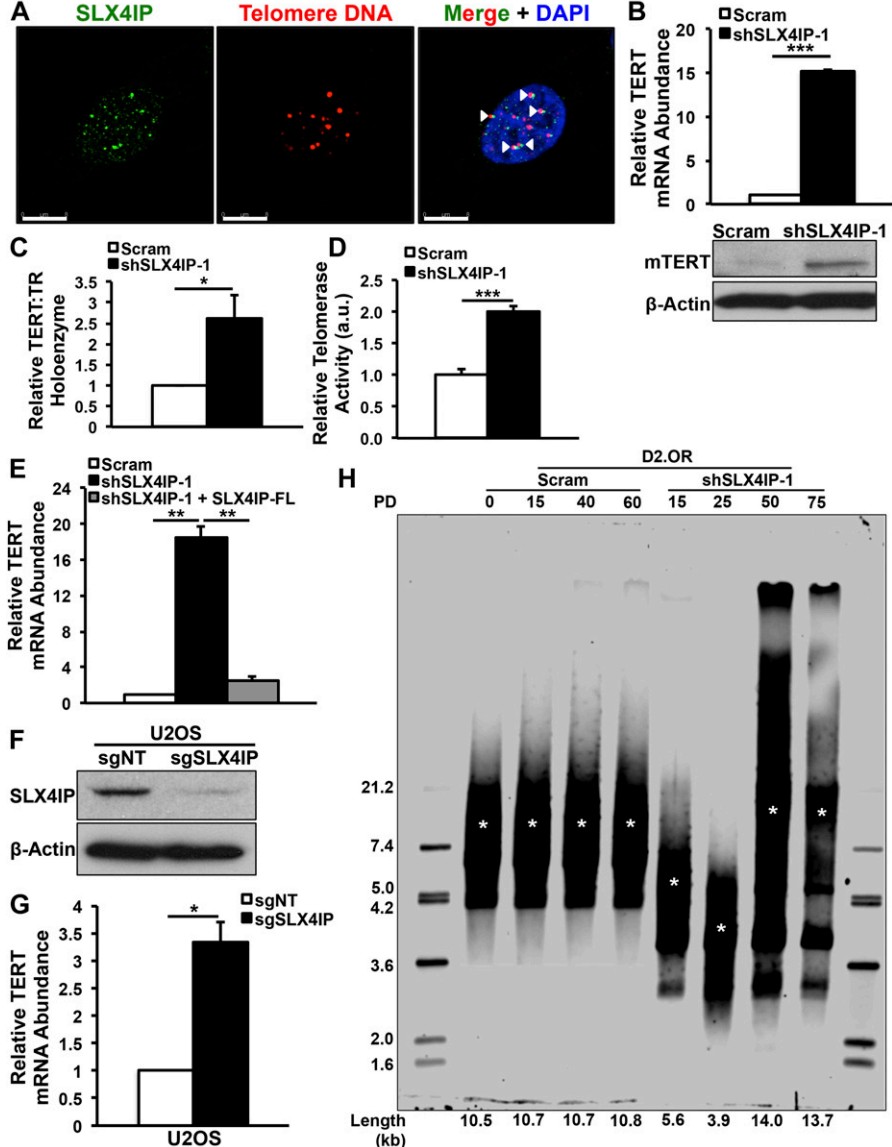

**Figure 2. Altered SLX4IP expression is associated with telomerase activation.**
**(A)** Representative IF/FISH images showing SLX4IP localization to telomeres (arrowheads) in D2.OR cells. Scale bar: 8 $\mu$m. **(B)** *Top*: qRT-PCR of TERT mRNA in the indicated D2.OR derivatives. *Bottom*: Immunoblot image of TERT protein abundance. $n = 3$ for qRT-PCR. **(C)** qRT-PCR of mature telomerase core holoenzyme after RNA immunoprecipitation of TERT-bound TR in specified D2.OR derivatives ($n = 3$). **(D)** Quantitation of telomerase enzyme activity in parental and SLX4IP-depleted D2.OR cells ($n = 5$ replicates per cell line). **(E)** qRT-PCR of TERT mRNA in parental, SLX4IP-depleted, and SLX4IP-reconstituted D2.OR derivatives ($n = 3$). **(F)** Representative immunoblot confirming SLX4IP knockout in U2OS cells. **(G)** qRT-PCR of TERT mRNA in parental and SLX4IP-null U2OS cells ($n = 3$). **(H)** Telomere restriction fragment Southern blot quantifying telomere length in parental and SLX4IP-depleted D2.OR derivatives at four distinct population doublings (PDs). Length refers to average telomere length in each lane (horizontal) or reference band size (vertical). Asterisks indicate average telomere length in each lane. **(B, C, D, E, G)** *$P < 0.05$, **$P < 0.01$, ***$P < 0.001$, Mann–Whitney $U$ test (Panels B, C, D, and G) or Kruskal–Wallis test (Panel E).

## Inverse SLX4IP and TERT expression patterns correlate with breast cancer subtypes and clinical outcomes

Human and mouse telomere dynamics differ widely (Calado & Dumitriu, 2013), raising the possibility that the relationship between SLX4IP and telomere maintenance observed in D2.OR cells may not be conserved in humans. To address this important question, we used a diverse array of primary breast cell and patient-derived tissue sources to evaluate the relationship between SLX4IP and TERT in human breast cancers. Examination of SLX4IP and TERT expression in patient-derived xenograft specimens (Zhang et al, 2013) revealed an inverse correlation between the expression of these two genes. Indeed, the directionality of this correlation was distinct in triple-negative breast cancer (TNBC) versus HER2-enriched (HER2$^+$) breast cancer, such that TNBCs exhibited a SLX4IP$^{Low}$TERT$^{High}$ expression profile (Fig 5A) as compared with the SLX4IP$^{High}$/TERT$^{Low}$ pattern observed in HER2$^+$ breast cancers

(Fig 5B). Importantly, this subtype-dependent trend in SLX4IP expression was mirrored in a larger breast cancer cohort housed within The Cancer Genome Atlas (Fig 5C; [Gao et al, 2013]). These findings indicate that SLX4IP and TERT display inverse expression patterns across genetically distinct human breast cancers, an event that evinces subtype-specific regulation of telomere homeostasis.

We next assessed the connection between the inverse expression patterns of SLX4IP and TERT and breast cancer patient outcomes, including progression to metastasis. Pairwise analysis of primary breast tumors with matched central nervous system (CNS) metastases uncovered striking patterns of SLX4IP and TERT expression. As shown in Fig 5D, the inverse relationship between SLX4IP and TERT was conserved in both TNBC and HER2$^+$ breast cancers, and these aberrations manifested specifically during metastasis to the CNS. Along the same lines, recurrence-free survival of TNBC patients was substantially reduced in those individuals with low

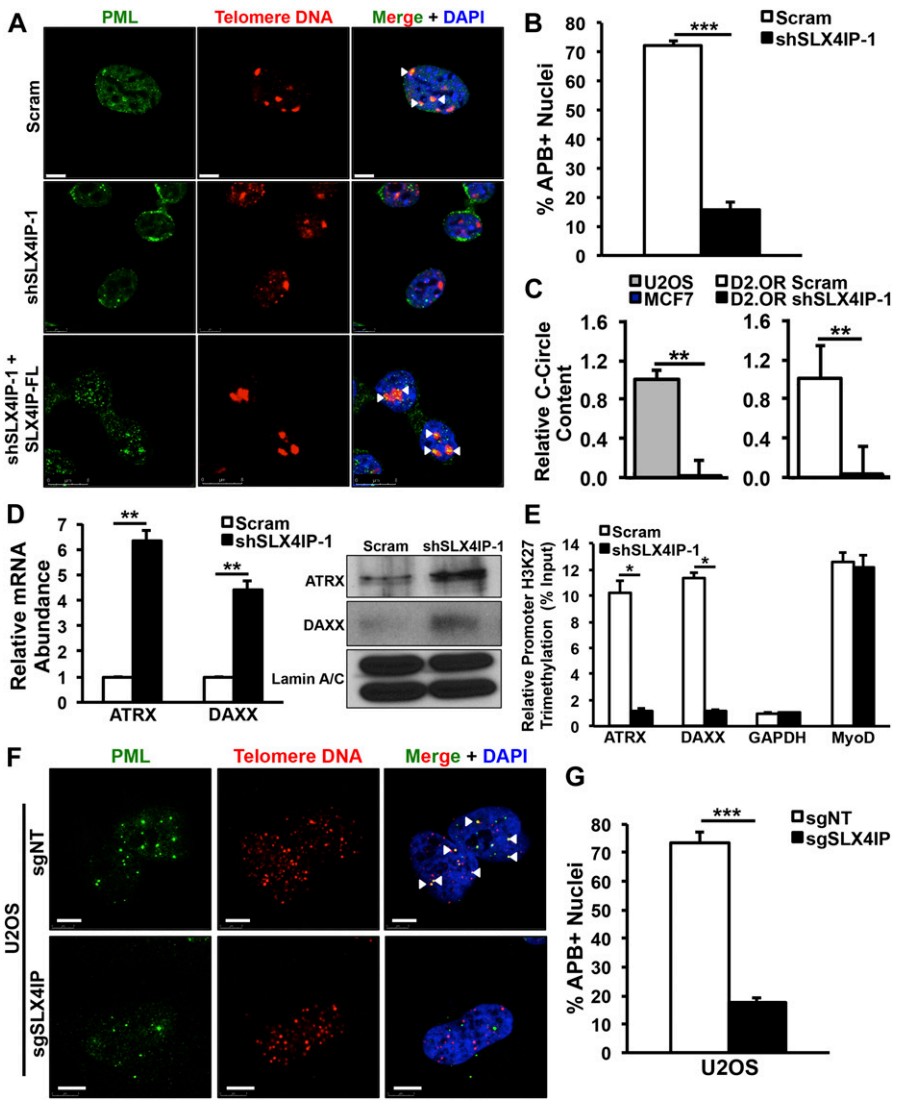

**Figure 3.  SLX4IP facilitates switching between alternative lengthening of telomere and telomerase for telomere maintenance.**
**(A)** Representative IF/FISH images illustrating the presence of APBs (arrowheads) in parental D2.OR cells and loss of these structures after SLX4IP depletion. In addition, cells re-acquire APBs upon rescue of SLX4IP expression. Scale bar: 5 μm. **(B)** APB quantification in parental (*n* = 213) and SLX4IP-depleted (*n* = 182) D2.OR cells, as a percentage of total nuclei observed. **(C)** *Left*: qRT-PCR of C-circle abundance in U2OS (alternative lengthening of telomere-positive) and MCF7 (telomerase-positive) control cell lines. Data are normalized to U2OS C-circle content (*n* = 3). *Right*: qRT-PCR of C-circle abundance in the indicated D2.OR derivatives. Data are normalized to parental C-circle content (*n* = 3). **(D)** *Left*: qRT-PCR of ATRX and DAXX mRNA in parental and SLX4IP-depleted D2.OR cells. *Right*: representative Western blot image of ATRX and DAXX protein expression. **(E)** H3K27me3 ChIP demonstrating epigenetic silencing of ATRX and DAXX in parental D2.OR cells. GAPDH serves as an actively transcribed gene and MyoD serves as a silenced gene (*n* = 3). **(F)** Representative IF/FISH images showing loss of APBs in SLX4IP-null compared with parental U2OS cells. Scale bar: 5 μm. **(G)** APB quantification in parental (*n* = 115) and SLX4IP-depleted (*n* = 110) U2OS derivatives, as a percentage of total nuclei observed (*n* = 3). **(B, C, D, E, G)** *P < 0.05, **P < 0.01, ***P < 0.001, Mann–Whitney *U* test (Panels B, D, E, and G) or Kruskal–Wallis test (Panel C).

SLX4IP or high TERT expression (i.e., SLX4IP[Low]/TERT[High]; Fig 5E), with the opposite expression profile holding similar prognostic value for HER2[+] breast cancer patients (i.e., SLX4IP[High]/TERT[Low]; Fig 5F). Taken together, these results reveal that SLX4IP and TERT are intricately related to one another in distinct human breast cancer subtypes, and this relationship presages metastatic progression and patient survival.

## TMMs can be therapeutically targeted in metastatic breast cancer cells

Given the observed association between TMM selection and metastatic behavior, and the fact that metastatic progression is a major determinant of breast cancer patient outcomes (Mariotto et al, 2017), we set out to define the efficacy of small molecules that selectively target each TMM as a means to eradicate metastatic breast cancer cells. Although the general susceptibility of ALT-driven tumors to ataxia-telangectasia and Rad3-related (ATR) inhibition

remains an open question (Flynn et al, 2015; Deeg et al, 2016), recent evidence demonstrated that the growth of some ALT[+] tumors is prevented by inactivation of ATR (Charrier et al, 2011; Foote et al, 2013; Flynn et al, 2015), which is a DNA damage–responsive kinase that stabilizes transient single-stranded DNA intermediates during homologous recombination (Sorensen et al, 2005). Accordingly, Fig S4A shows that administration of the ATR inhibitor, AZ20, blocked ATR-mediated phosphorylation of the DNA damage kinase Chk1 in response to the replication fork stalling without impacting the phosphorylation of ribosomal S6K in response to insulin (Fig S4B). Importantly, administering AZ20 to ALT[+] D2.OR cells dramatically reduced their 3D-outgrowth (Fig 6A), as did administration of the ATR inhibitor, VE-821 (Fig 6B; [Charrier et al, 2011]) and the BLM RecQ helicase inhibitor, ML216 (Fig 6B; [Rosenthal et al, 2010; Nguyen et al, 2013]). Interestingly, the antiproliferative activities of AZ20 failed to occur in SLX4IP-deficient D2.OR cells regardless of their TERT expression status (Figs 6A and C and S5A). AZ20 also induced cytotoxicity and apoptosis in ALT[+] tumor cells derived from mice (D2.OR;

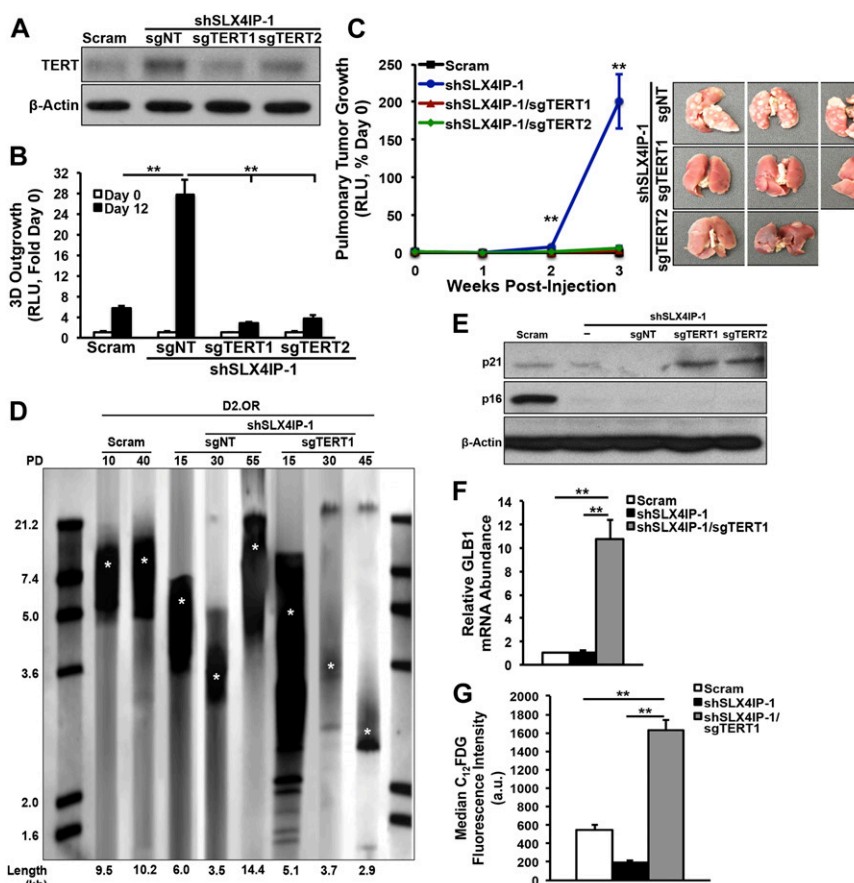

Figure 4. Telomerase is required for the acquisition of metastatic features after SLX4IP inactivation.
(A) Representative immunoblot confirming CRISPR-mediated reduction of TERT expression in SLX4IP-depleted D2.OR derivatives. (B) Quantitation of 3D-outgrowth of parental, and TERT[+] and TERT[−]/SLX4IP-depleted D2.OR cells (n = 4). (C) Left: bioluminescence imaging of pulmonary tumor formation in mice inoculated with denoted D2.OR derivatives. Asterisks indicate significant differences in tumor formation between SLX4IP-depleted and both TERT knockout cell lines at a given time point. Right: lungs harvested from mice inoculated with noted D2.OR derivatives (n = 5). (D) Telomere restriction fragment Southern blot quantifying telomere length in parental D2.OR cells, as well as SLX4IP-depleted derivatives with or without TERT expression, at the indicated population doublings (PDs). Asterisks indicate average telomere length in each lane. (E) Representative immunoblot showing up-regulation of p21 after loss of TERT in SLX4IP-depleted D2.OR cells. (F) qRT-PCR of senescence-associated $\beta$-galactosidase (SA-$\beta$-gal; GLB1) mRNA in D2.OR cells with indicated genotypes (n = 3). (G) Median fluorescence intensity values obtained by flow cytometric analysis of 5-dodecanoylaminofluorescein di-$\beta$-D-galactopyranoside (C$_{12}$FDG) metabolism by specified D2.OR cell lines. C$_{12}$FDG is a fluorescent substrate for SA-$\beta$-gal (n = 3). **$P < 0.01$, Kruskal–Wallis test.

Fig 6D) and humans (U2OS; Fig S5B). Along these lines, pharmacological inhibition of ATR significantly reduced the quantity of APBs in parental D2.OR cells (Fig 6E), a finding consistent with previous observations that ATR inhibition attenuates C-circle formation in ALT[+] cells (Flynn et al, 2015). Collectively, these findings suggest that SLX4IP expression and its association with ALT phenotypes influence cancer cell susceptibility to the anticancer activities of ATR and BLM inhibitors.

We recently identified the pyrimidine analog 5-fluoro-2′-deoxyuridine (5-FdU; also known as floxuridine) as a novel nucleoside substrate for telomerase, resulting in a replication protein A-dependent DDR and telomeric catastrophe in telomerase-positive cancer cells (Zeng et al, 2018). We found that replication protein A was indeed phosphorylated robustly in SLX4IP-deficient D2.OR (i.e., SLX4IP[Low]TERT[High]) cells treated with 5-FdU (Fig S4C). Likewise, the longitudinal 3D-outgrowth of SLX4IP-deficient D2.OR cells was exquisitely sensitive to treatment with 5-FdU, which contrasted sharply with the inherent resistance to 5-FdU exhibited by their parental (i.e., SLX4IP[High]TERT[Low]) and TMM-deficient D2.OR counterparts (Figs 6F and G and S5C). The anticancer activities of 5-FdU triggered the formation of telomere dysfunction-induced foci (TIFs), specifically in SLX4IP[Low]/TERT[High] D2.OR cells, pointing to the induction of telomeric DNA damage (Figs 6I and S5D). Notably, the related compound 5-fluorouracil (5-FU), which does not serve as a telomerase substrate (Zeng et al, 2018), failed to impact the growth of either parental or SLX4IP-deficient D2.OR cells

(Fig 6F). Finally, the relationship between SLX4IP and TERT expression in dictating breast cancer cell sensitivity also extends to the members of the murine 4T1 breast cancer progression series, which comprises (i) 67NR cells, which are weakly tumorigenic and nonmetastatic, (ii) 4T07 cells, which are systemically invasive, and (iii) 4T1 cells, which are highly metastatic (Aslakson & Miller, 1992). Importantly, metastatic 4T07 and 4T1 cells possessed a SLX4IP[Low]/TERT[High] gene expression profile and exhibited extreme sensitivity to 5-FdU administration, a response that contrasted sharply with the resistance to 5-FdU exhibited by SLX4IP[High]TERT[Low] 67NR cells (Fig S5E–G). Taken together, these findings indicate that SLX4IP and TMM identity are strongly associated with disease progression and therapeutic response to ATR inhibitors and 5-FdU; they also suggest that elucidating breast cancer TMM status may provide new insights to inform treatment selection.

## SLX4IP correlates with telomere homeostasis and therapeutic response in human breast cancer

We next determined whether our murine-based SLX4IP findings could be generalized to human breast cancer models, and more importantly, whether TMMs can also be targeted therapeutically in these systems. Consistent with the observation that aggressive HER2[+] breast cancers and TNBCs are associated with disparate patterns of SLX4IP and TERT expression (Fig 5), we found that

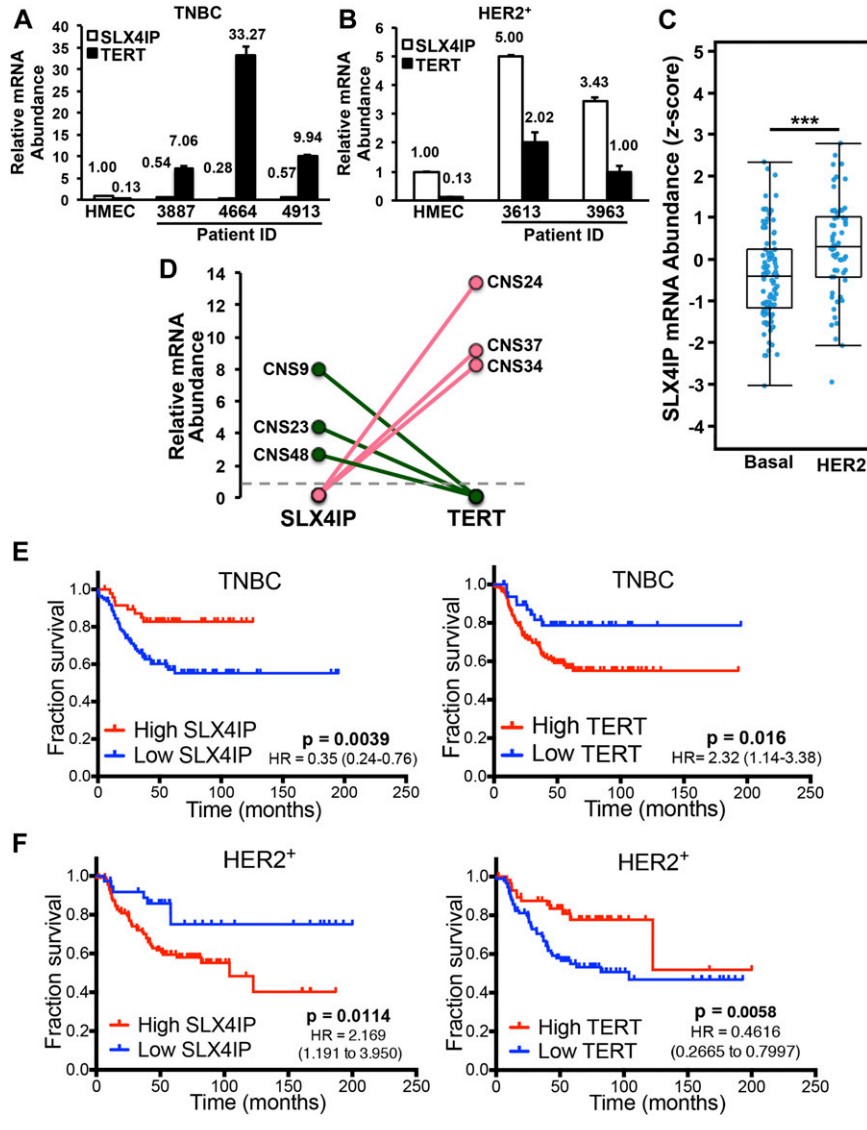

**Figure 5. Inverse SLX4IP and TERT expression patterns correlate with breast cancer subtypes and clinical outcomes.**
**(A, B)** qRT-PCR of SLX4IP and TERT mRNA in triple-negative (A) and HER2-enriched (B) patient-derived xenograft cohorts. Each patient was assigned a unique numerical identifier. HMEC, human mammary epithelial cells (*n* = 3). **(C)** Boxplot displaying SLX4IP expression (represented by microarray z-score) stratified by breast cancer subtype. n = 95 basal and 58 HER2-enriched samples. **(D)** Dot plot showing SLX4IP and TERT expression in central nervous system metastases (normalized to matched primary tumor specimens) from patients with triple-negative (pink) or HER2-enriched (green) breast cancer. The dotted line denotes no change in gene expression between a primary tumor and its matched metastasis. **(E, F)** Kaplan–Meier plots showing recurrence-free survival of patients with triple-negative (E) or HER2-enriched (F) breast cancers, stratified by SLX4IP (*left*) or TERT (*right*) expression. Numbers in parentheses show 95% confidence intervals. In **(E, F)**, significance was determined using a log-rank (Mantel–Cox) test. ***P < 0.001, Mann–Whitney *U* test. HR, hazard ratio.

different human breast cancer cell lines also show varying patterns of SLX4IP and TERT expression (Fig 7A), suggesting that the interplay between SLX4IP and TERT expression in human breast cancers is linked to TMM identity and drug sensitivity in a manner reminiscent of D2.OR cells. To test this supposition, we rendered HER2⁺ BT474 cells (i.e., SLX4IP^High^TERT^Low^; Fig 7A) deficient in SLX4IP expression using CRISPR/Cas9, and conversely, engineered triple-negative HCC1806 cells (*i.e.*, SLX4IP^Low^TERT^High^; Fig 7A) to overexpress SLX4IP by lentiviral transduction (Fig 7B). At baseline, APBs were readily apparent in BT474 cells (Fig 7C and D), and the 3D-outgrowth of these cells was strongly suppressed by AZ20 (Fig 7E), findings consistent with the notion that BT474 cells rely upon ALT. Accordingly, rendering BT474 cells deficient in SLX4IP expression dramatically reduced their abundance of APBs and instituted resistance to ATR inhibition (Fig 7C–E). Importantly, the cellular features of ALT and sensitivity to AZ20 were reinstated in BT474 cells after reconstitution of SLX4IP expression using murine SLX4IP, which was not targeted by our human sgSLX4IP construct (Fig 7F and G).

In stark contrast, ectopic expression of SLX4IP in HCC1806 cells produced a concomitant down-regulation of TERT (Fig 7H) and acquisition of resistance to 5-FdU (Fig 7I). Last, we selected representative SLX4IP^Low^/TERT^High^ and SLX4IP^High^/TERT^Low^ TNBC cell lines and measured their 3D-outgrowth in the absence or presence of 5-FdU with or without the addition of the telomerase inhibitor BIBR1532. The outgrowth of SLX4IP^Low^/TERT^High^ HCC1806 and HCC1143 cells was readily inhibited by administration of 5-FdU, whereas SLX4IP^High^/TERT^Low^ Hs578T cells were unaffected by this treatment regimen. Moreover, the effect of 5-FdU was blocked when the enzymatic activity of telomerase was inhibited by addition of BIBR1532 (Fig S6A and B), suggesting the cytotoxic effects of 5-FdU were in fact mediated by telomerase. Taken together, these findings support the concept that the interplay between SLX4IP and TERT dictates TMM selection and drug sensitivity, such that (i) SLX4IP^Low^/TERT^High^ breast cancer cells succumb to 5-FdU treatment and resist ATR inhibition, and (ii) SLX4IP^High^/TERT^Low^ breast cancer cells succumb to ATR inhibition and resist 5-FdU treatment.

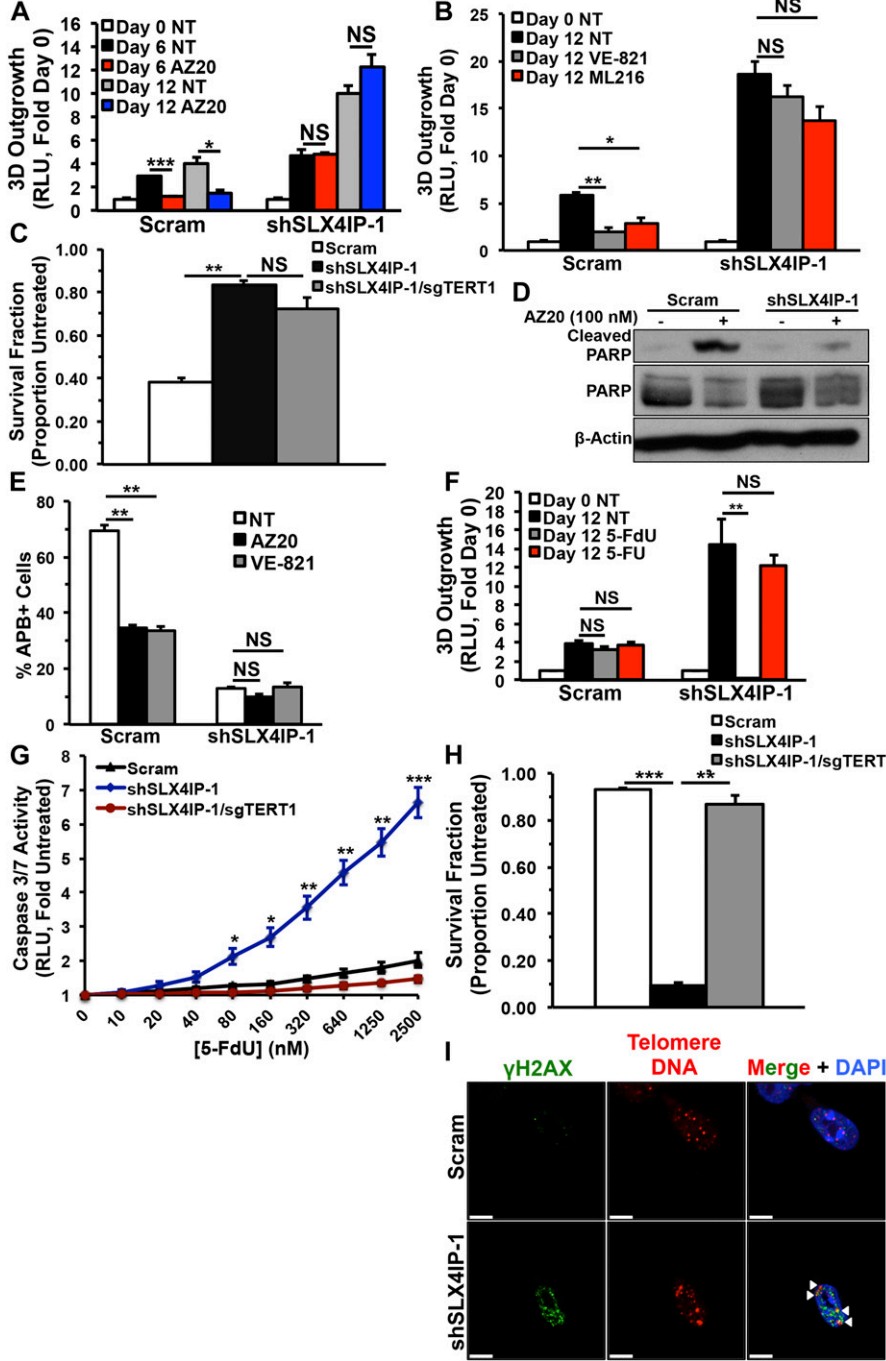

**Figure 6. Telomere maintenance mechanisms can be therapeutically targeted in metastatic breast cancer cells.**
**(A)** Quantitation of 3D-outgrowth of parental and SLX4IP-depleted D2.OR cells treated with 100 nM AZ20 or diluent (NT; $n = 4$). **(B)** Quantitation of 3D-outgrowth of parental and SLX4IP-depleted D2.OR cells treated with VE-821 (1 $\mu$M), ML216 (3 $\mu$M), or diluent ($n = 4$). **(C)** Survival quantification of colonies formed by specified D2.OR derivatives treated with AZ20 (compared with untreated, which has a survival fraction of 1.0; $n = 3$). **(D)** Representative immunoblot showing poly(ADP-ribose) polymerase (PARP) cleavage in parental and SLX4IP-depleted D2.OR cells after AZ20 exposure. PARP cleavage serves as a marker of cell death. **(E)** APB quantification in parental and SLX4IP-depleted D2.OR cells treated with either diluent (PBS), AZ20 (100 nM), or VE-821 (1 $\mu$M) as indicated, with the percentage of total nuclei observed being presented. **(F)** Quantitation of 3D-outgrowth of parental and SLX4IP-depleted D2.OR cells treated with 250 nM of either 5-FdU, 5-FU, or diluent ($n = 4$). **(G)** Quantitation of pro-apoptotic caspase 3/7 activity in D2.OR derivatives treated with varying doses of 5-FdU ($n = 3$). **(H)** Survival quantification of colonies formed by specified D2.OR derivatives treated with 5-FdU ($n = 3$). **(I)** Representative IF/FISH images showing TIF formation (arrowheads) in SLX4IP-depleted D2.OR cells, but not in their parental counterparts treated with 5-FdU (250 nM). TIFs are defined by co-localization of telomere DNA and the DNA damage-specific histone variant $\gamma$H2AX. Scale bar: 5 $\mu$m. **(A, B, C, E, F, G, H)** *$P < 0.05$, **$P < 0.01$, ***$P < 0.001$, Mann–Whitney $U$ test (Panel A) or Kruskal–Wallis test (Panels B, C, E, F, G, and H). NS, not significant.

## Administration of 5-FdU eradicates telomerase-dependent metastasis formation and promotes emergence of ALT

Spurred by our in vitro findings, we next investigated the effectiveness of 5-FdU to eliminate pulmonary tumors formed by SLX4IP-deficient D2.OR cells (i.e., SLX4IP[Low]/TERT[High]) in syngeneic BALB/c mice. Fig 8A provides a schematic overview of our preclinical study design, which also included a 5-FU treatment group based on reports that 5-FdU and 5-FU have overlapping mechanisms of action via conversion to a common metabolite (Malet-Martino & Martino, 2002). As depicted in Fig 8B and C, mice inoculated intravenously with SLX4IP-deficient D2.OR cells and treated with two different concentrations of 5-FdU showed no evidence of pulmonary tumor formation, a result that contrasted sharply with the high-grade tumors present in the lungs of mice treated with either diluent (i.e., PBS) or 5-FU. Moreover, mice that developed high tumor burdens showed marked reduction in body weight, a phenomenon that was not observed in those treated with 5-FdU (Fig S7A). Although

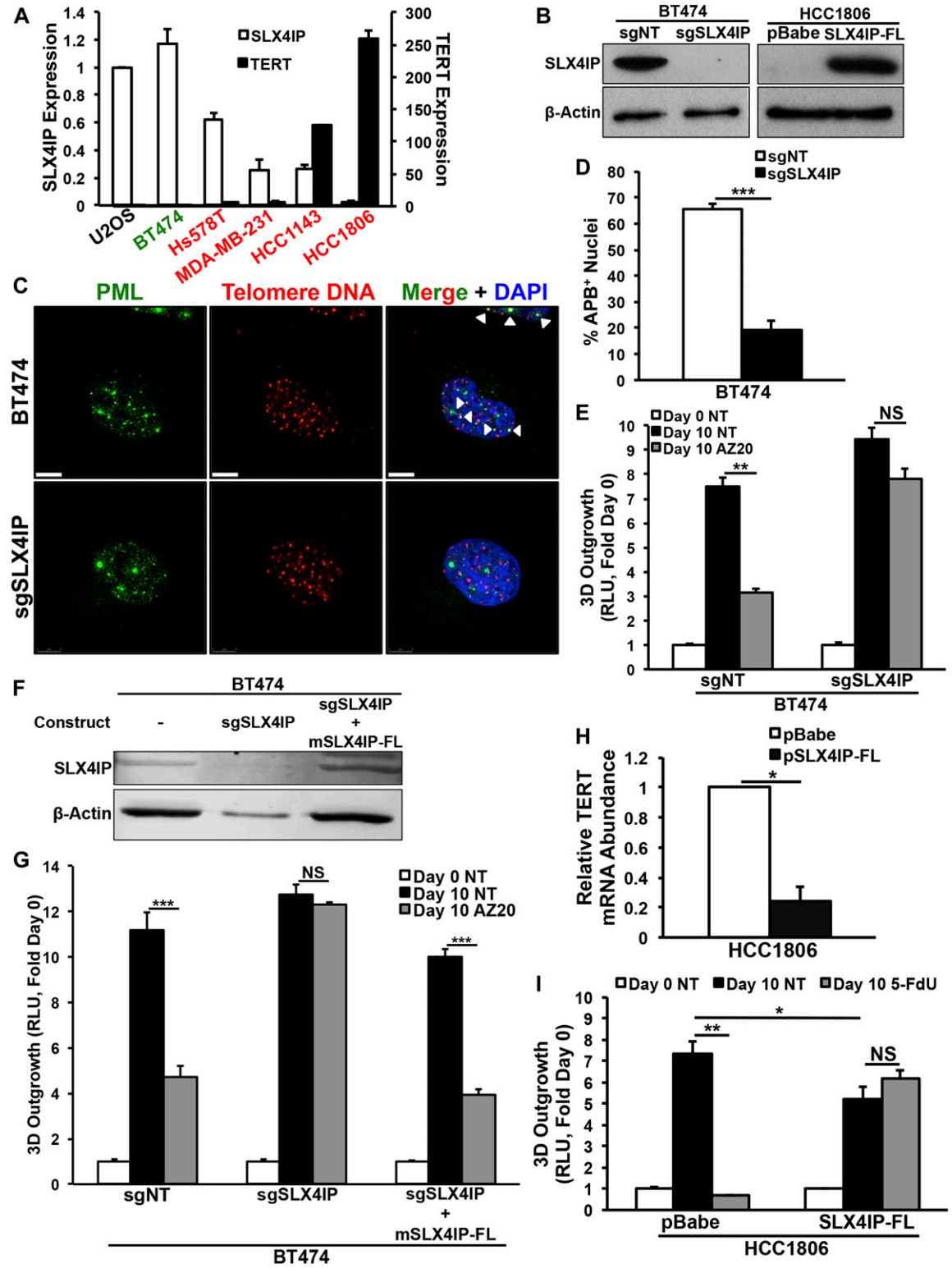

**Figure 7. SLX4IP correlates with telomere homeostasis and therapeutic response in human breast cancer.**
**(A)** qRT-PCR of SLX4IP and TERT mRNA in cell lines used in this figure. Axes display relative abundance of the indicated transcript. HER2-enriched lines are shown in green, and triple-negative breast cancer lines are shown in red (*n* = 3). **(B)** Representative immunoblots confirming SLX4IP knockout (*left*) or ectopic expression (*right*) in BT474 and HCC1806 cells, respectively. **(C)** Characteristic IF/FISH images illustrating presence of APBs (arrowheads) in BT474 cells and loss of these structures after SLX4IP knockout. Scale bar: 5 *μ*m. **(D)** APB quantification in parental (*n* = 104) and SLX4IP-null (*n* = 124) BT474 cells, as a percentage of total nuclei observed. **(E)** Quantitation of 3D-outgrowth of parental and SLX4IP-null BT474 cells treated with AZ20 (150 nM) or diluent (NT; *n* = 4). **(F)** Representative immunoblot confirming ectopic expression of full-length murine SLX4IP (mSLX4IP-FL) in SLX4IP-null BT474 cells. **(G)** Quantitation of 3D-outgrowth of parental, SLX4IP-null, and SLX4IP-reconstituted BT474 cells treated

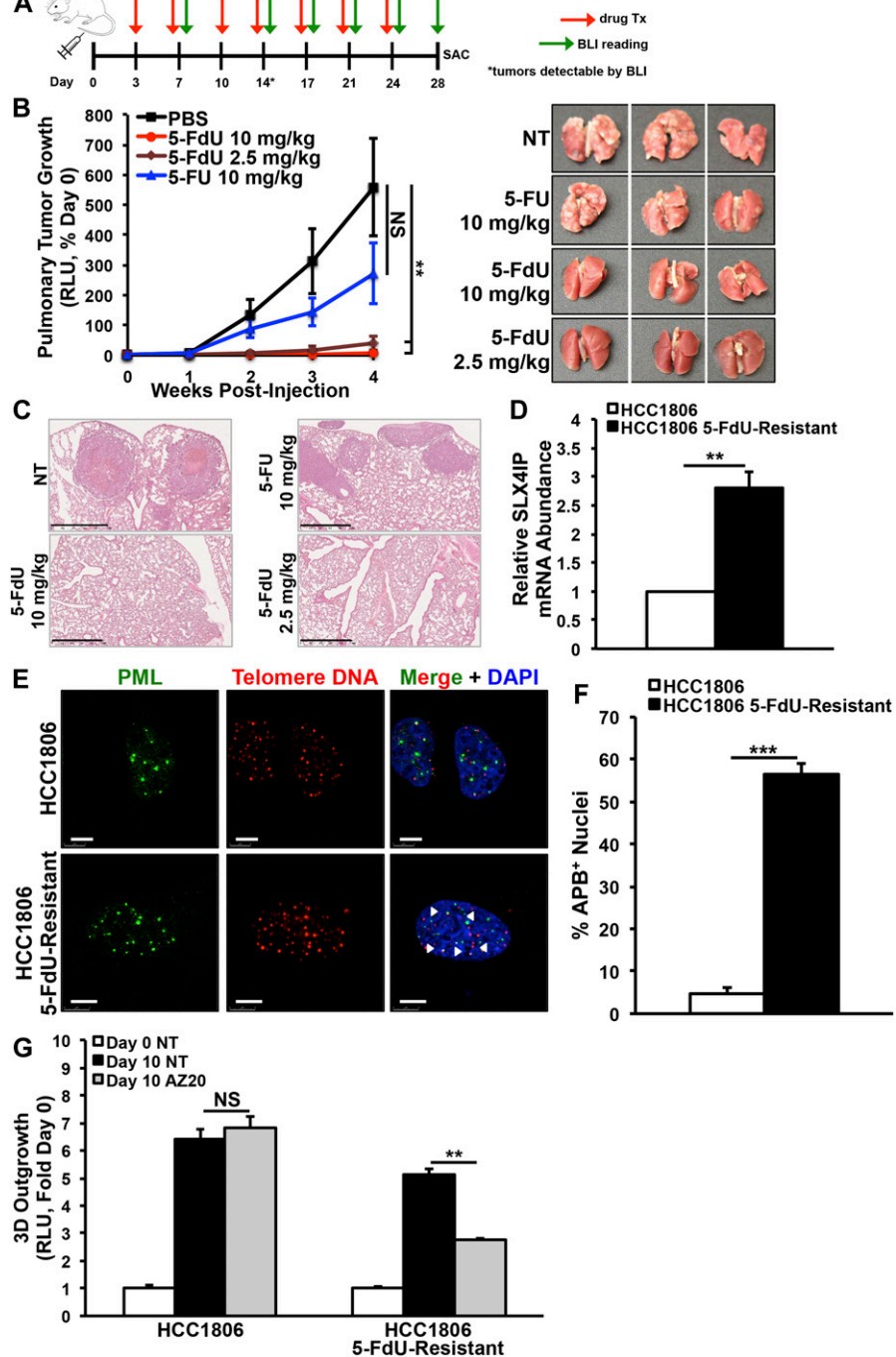

**Figure 8. Administration of 5-FdU eradicates telomerase-dependent metastasis formation and promotes emergence of alternative lengthening of telomere.**
**(A)** Schematic overview depicting BLI and drug administration schedule in BALB/c mice. **(B)** *Left*: BLI of pulmonary tumor formation in mice inoculated with SLX4IP-depleted D2.OR cells and treated with the indicated drugs or diluent (PBS). *Right*: lungs harvested from treated and untreated mice, showing absence of overt tumors in mice treated with 5-FdU, whereas 5-FU exerted minimal effect (*n* = 5). **(C)** Representative H&E–stained sections of lungs harvested from mice receiving the specified treatments. Scale bar: 1 mm. **(D)** qRT-PCR of SLX4IP mRNA in 5-FdU–sensitive and 5-FdU–resistant HCC1806 cells (*n* = 3). **(E)** Representative IF/FISH images showing acquisition of APBs in 5-FdU–resistant HCC1806 cells. Scale bar: 5 μm. **(F)** APB quantification in parental (*n* = 170) and 5-FdU–resistant (*n* = 140) HCC1806 derivatives, as a percentage of total nuclei observed. **(G)** Quantitation of 3D-outgrowth of 5-FdU–sensitive and 5-FdU–resistant HCC1806 cells treated with AZ20 (100 nM) or diluent (*n* = 4). **(B, D, F, G)** **P < 0.01, ***P < 0.001, Mann–Whitney *U* test (Panels D and F) or Kruskal–Wallis test (Panels B and G). NS, not significant.

rare and difficult to detect, micrometastases that did arise in 5-FdU–treated mice showed signs of DNA damage accumulation (Fig S7B). These important findings support the notion that administration of 5-FdU to telomerase-driven breast tumors initiates a DDR that prevents their metastatic outgrowth and recurrence.

Finally, one of the major clinical challenges associated with pharmacologic targeting of telomerase is the development of therapeutic resistance, including the selection and/or adaptation of a cell population that relies upon ALT. Although this phenomenon has been observed after treatment with telomerase inhibitors

with AZ20 (150 nM) or diluent (NT; *n* = 4). **(H)** qRT-PCR of TERT mRNA in HCC1806 cells exhibiting wild-type or overexpression of SLX4IP (*n* = 3). **(I)** Quantitation of 3D-outgrowth of parental and SLX4IP-overexpressing HCC1806 cells treated with 5-FdU (500 nM) or diluent (*n* = 4). **(D, E, G, H, I)** *P < 0.05, **P < 0.01, ***P < 0.001, Mann–Whitney *U* test (Panels D and H) or Kruskal–Wallis test (Panels E, G, and I). NS, not significant.

(Hu et al, 2012), it remains to be determined whether TMM-targeting agents (e.g., 5-FdU) are subjected to similar resistance mechanisms. Intriguingly, long-term treatment of parental HCC1806 cells with 5-FdU yielded a chemoresistant subpopulation. Indeed, Fig 8D shows that these emergent cells contained dramatically up-regulated levels of SLX4IP expression; they also possessed elevated features of ALT, including increased APBs (Fig 8E and F) and AZ20 sensitivity (Fig 8G). Collectively, these findings support a role for SLX4IP in mediating ALT and reinforce the notion of deploying combinatorial approaches to target TMMs in breast cancers.

## Discussion

This study elucidates the utility of SLX4IP as a potential predictive marker of breast cancer metastasis and patient survival and reveals its close connection with telomere homeostatic pathways. Molecular indicators of these pathways, in turn, also convey critical cancer prognostic and therapeutic information that can provide immense clinical insight. There is much yet to be determined about the molecular features of SLX4IP that may enable its contributions to TMMs, particularly via the ALT pathway. As a member of the SLX4 SSE, SLX4IP may modulate the activity of DNA repair nucleases, such as SLX1, MUS81, or XPF (ERCC4) (Zhang et al, 2019), or the mismatch repair complex, MSH2–MSH3 (Svendsen et al, 2009). The SLX4 SSE possesses affinity for a wide spectrum of DNA structures, including Holliday junctions (Svendsen et al, 2009) and telomeric joint molecules (Sarkar et al, 2015). Thus, SLX4IP may serve a broad role in maintaining genomic integrity by resolving telomeric DNA damage and avoiding telomere crisis by maintaining telomeres above their critical length.

ALT occurs after telomere attrition or deprotection, at which point, the cell interprets telomeres as DNA double-strand breaks. This activates the homology-directed repair (HDR) pathway, resulting in the synthesis of new telomeric DNA in a manner similar to break-induced replication that occurs in response to single-ended double-strand breaks formed at stalled replication forks (Pickett & Reddel, 2015). In addition to acting as a scaffold for nuclease assembly, SLX4 binds to (i) telomeric repeat-binding factor 2 (TRF2) and TRF2-interacting protein (TERF2IP, also known as RAP1), both of which are members of the shelterin complex and prevent aberrant telomere DDRs (Svendsen et al, 2009). Interestingly, RAP1 has been shown to block HDR at telomeres in part by repressing SLX4 localization (Rai et al, 2016). Moreover, depletion of RAP1 or expression of TRF2 mutants missing the RAP1-binding domain stimulates telomere sister chromatid exchange, a hallmark of ALT (Bailey et al, 2004; Sfeir et al, 2010). SLX4 has generally been found to have a negative effect on telomere length, including in the context of HDR by the action of SLX1 (Wan et al, 2013). However, MUS81 endonuclease activity is required for telomere recombination and ALT⁺ cell survival (Zeng et al, 2009). Thus, the functions of SLX4 and its associated proteins in telomere maintenance remain incompletely understood. Indeed, the bifurcation between telomere lengthening and shortening secondary to homologous recombination is controlled in part by the interplay between the activities of the SLX4 SSE and BLM (Sobinoff et al, 2017). Conceivably, SLX4IP may regulate ALT by coordinating RAP1–telomere interactions,

or by recruiting or activating SLX4, SLX1, MUS81, or BLM. Indeed, an interaction between SLX4IP and BLM has recently been reported, suggesting that SLX4IP acts as a negative regulator of BLM activity and ALT (Panier et al, 2019). Given the discrepancy between these findings and those presented herein, future studies are clearly warranted to more fully understand the functions of SLX4IP in regulating ALT and in coordinating TMMs.

TMM identity is classically viewed as a stable property of immortalized cells. More accurately, immortalization achieved during neoplastic transformation is generally presumed to be carried out by telomerase, with ALT serving as a reserve mechanism that becomes operational when telomerase function is disrupted. More recent examinations have uncovered pathologic evidence of ALT in ~15% of cancers, most frequently in tumors of mesenchymal origin, such as osteosarcomas and gliomas. Notably, however, ALT has also been detected in cancers of the bladder, cervix, endometrium, esophagus, kidney, liver, and lung, and in non-glioma CNS tumors (Heaphy et al, 2011b). In line with these observations, evidence of ALT is observed in a subset (~15%) of HER2-enriched breast cancer patients who presented with lymph node metastases at the time of initial diagnosis and ultimately succumbed to highly aggressive disease (Subhawong et al, 2009). In contrast, no evidence of ALT was found in TNBC patients, suggesting that the telomeres in these tumors were maintained by telomerase. These findings are consistent with our assertion that a SLX4IP^High/TERT^Low gene expression profile is indicative of ALT and associated with poor outcomes specifically in HER2-enriched breast cancer patients (Fig 5). Our investigation also asserts the possible existence of an innate plasticity in TMM selection. Moreover, the relationship between SLX4IP and TMM acquisition likely represents only one facet of a complex regulatory network that receives inputs from a multitude of cell-intrinsic and microenvironmental cues, events likely to be honed by signaling inputs derived from Wnt/β-catenin, NF-κB (Yin et al, 2000), and c-Myc (Wu et al, 1999). As such, future studies need to explore (i) the significance of SLX4IP and ALT as a driver of tumor progression, (ii) the plasticity inherent in the establishment and preservation of TMM identity, and (iii) the regulatory landscape of SLX4IP and its connections to the signaling pathways listed above.

The presence of active telomerase in many cancers makes this enzyme an attractive target for therapies that disrupt cancer cell function while leaving untransformed cells intact. Indeed, telomerase inhibitors have seen success in preclinical models of lung, breast, and pancreatic cancers, as well as in myeloid leukemia (Dikmen et al, 2005; Joseph et al, 2010; Bruedigam et al, 2014). Unfortunately, these agents have exhibited little-to-no survival benefit in clinical trials, including those involving breast cancer patients (Xu & Goldkorn, 2016). Multiple mechanisms have been proposed for the failure of telomerase inhibitors in these settings. First, by inhibiting telomerase, these drugs provoke telomere attrition culminating in senescence or apoptosis. However, this process requires numerous rounds of cell division for telomeres to reach their critical length. This lag time provides an opportunity for cancer cells to adopt resistance mechanisms, including ALT (Hu et al, 2012). In addition, a subpopulation of therapy-induced senescent cells may persist as chemoresistant clones harboring acquired stem-like features, ultimately leading to disease recurrence (Milanovic et al, 2018). Nevertheless, targeting TMMs remains an

appealing therapeutic strategy in need of novel approaches, such as using nucleoside analogs that act as substrates for telomerase in a manner that is completely distinct from telomerase inhibition. Indeed, we found that 5-FdU co-opts telomerase activity to initiate cell death in telomerase-positive breast cancer cells and eradicate telomerase-driven metastatic disease (Figs 6–8). The beneficial effects of 5-FdU on metastatic breast cancer cells appear to be dependent upon telomerase, as evidenced by the loss of therapeutic efficacy after genetic ablation (Fig 6G and H) or pharmacologic inhibition (Fig S6A) of TERT. However, it remains possible that the 5-FdU is also misincorporated during break-induced telomere synthesis, an event that would conceivably produce deleterious effects on ALT cells. Although by no means definitive, this concern is partially mitigated by two important observations. First, 5-FdU does not inhibit the growth of U2OS cells (Fig S6A). Second, 5-FdU is incorporated into telomeres by telomerase when administered at doses below those at which it can be used by other DNA polymerases (Zeng et al, 2018). Thus, whereas future studies clearly need to examine the fate of 5-FdU in human breast cancers, our observations lend support to the notion that low-dose 5-FdU possesses high selectivity for telomerase and induces preferential cytotoxicity in telomerase-driven cancers. Interestingly, our work also reveals a pharmacodynamic divergence between 5-FdU and 5-FU (Figs 6F and 8B and C), thereby shedding new mechanistic light upon 5-FdU and its potential clinical repurposing toward novel targets such as TMMs. Our study further suggests that ALT may serve as an adaptive mechanism that is preferentially activated by dormant DTCs. Importantly, this may provide a unique targeting strategy for abrogating recurrent disease that can be accomplished through combinatorial targeting of multiple pathways (e.g., anti-ATR or anti-BLM in combination with standard-of-care or anti-telomerase agents). Experiments designed to test the therapeutic potential of combinatorial TMM-based treatments in preclinical therapy models are currently underway.

# Materials and Methods

### Cell lines

D2.OR and 4T1 progression series (67NR, 4T07, and 4T1) cells were obtained from Fred Miller (Wayne State University) and cultured in DMEM supplemented with 10% FBS. All other human breast cancer (MCF7, MDA-MB-231, BT474, Hs578T, HCC1143, and HCC1806) and ALT (U2OS) cell lines, and HEK293T cells were obtained from the American Type Culture Collection. Cells were cultured in DMEM supplemented with 10% FBS (MDA-MB-231, HEK293T), DMEM with 10% FBS and human recombinant insulin (0.01 mg/ml; MCF7 and Hs578T), McCoy's 5a media with 10% FBS (U2OS), RPMI-1640 medium with 10% FBS (HCC1143 and HCC1806), or Hybri-Care Medium with 10% FBS and sodium bicarbonate (1.5 g/l; BT474). All media were additionally supplemented with 1% penicillin–streptomycin (penstrep). All cells were grown in a 37°C incubator with 5% $CO_2$. Cell lines were engineered to stably express firefly luciferase by transfection with pNifty–CMV–luciferase followed by Zeocin selection (500 μg/ml). Chemoresistant HCC1806 derivatives were generated by treating HCC1806 cells with the nucleoside analog 5-fluoro-2'-deoxyuridine (5-FdU; Sigma-Aldrich) according to the following schedule: 3 d with 5-FdU → 2 d drug-free, 10 cycles, with stepwise increases in the concentration of 5-FdU from 100 nM → 3 μM.

### DNA constructs

SLX4IP knockdown was achieved by VSV-G lentiviral transduction of pLKO.1 containing either a nonspecific shRNA sequence or one of two gene-specific sequences (GE Dharmacon; Table S1). Telomerase-positive D2.OR cells were transduced with vectors harboring a chimeric single guide RNA scaffold (pLentiCRISPRv2) (Shalem et al, 2014), followed by selection with puromycin (5 μg/ml). Single guide RNA design was carried out using CHOPCHOP (Labun et al, 2016). Stable overexpression of SLX4IP was accomplished via transduction with pLenti CMV GFP expressing, FLAG-tagged, RNAi-resistant SLX4IP (pLenti-SLX4IP). SLX4IP cDNA was PCR-amplified using Platinum Pfx DNA Polymerase (Thermo Fisher Scientific) and purified using the QIAquick PCR Purification Kit (QIAGEN), digested with Sal I and EcoRV (New England Biolabs), and ligated into pENTR4-FLAG (QIAquick Gel Extraction Kit; QIAGEN). Generation of the overexpression construct was carried out using the Gateway LR Clonase II system (Thermo Fisher Scientific).

### VBIM and arbitrarily primed PCR

VBIM was adapted from existing screening platforms (Lu et al, 2009; Cipriano et al, 2012). Mutagenized cells were isolated by GFP FACS using a FACSAria II flow cytometer (BD Biosciences). To screen for metastatic mutants, VBIM-transduced cells were initially plated in three-dimension (3D) culture, and outgrowth-proficient clones were selected by light microscopic inspection, propagated, and injected intravenously into the lateral tail veins of 8-wk-old female BALB/c mice to assess pulmonary tumor formation. Gene identification was accomplished by amplifying VBIM-associated transcripts with nested VBIM-specific primers (forward) and a random hexamer primer containing a 5' recognition handle of known sequence (reverse; see Table S2). Arbitrarily primed PCR products were cloned into pGEM-T Easy (Promega), purified using the QIAprep Spin Miniprep Kit (QIAGEN), and subjected to Sanger sequencing.

### In vitro and in vivo bioluminescence monitoring

3D-outgrowth quantification and in vivo bioluminescence imaging were carried out as described (Gooding et al, 2017). Cells were cultured in appropriate media supplemented with 5% Cultrex, as well as 5-FdU or 5-fluorouracil (5-FU; Sigma-Aldrich), the ATR inhibitors AZ20 (4-{4-[(3R)-3-Methylmorpholin-4-yl]-6-[1-(methylsulfonyl)cyclopropyl]pyrimidin-2-yl]-1H-indole; MedChem Express) or VE-821 (3-amino-6-[4-(methlsulfonyl)phenyl]-N-phenyl-2-pyrazinecarboxamide; Sigma-Aldrich), the BLM inhibitor ML216 (1-(4-fluoro-3-(trifluoromethyl)phenyl)-3-(5-(pyridine-4-yl)-1,3,4-thiadiazol-2-yl)urea; MedChem Express), or the telomerase inhibitor BIBR1532 (Selleckchem) as indicated. For U2OS cells, Cultrex cushions were supplemented with type I collagen (3 mg/ml; BD Biosciences). In mice, 5-FdU and 5-FU were administered

by slow intravenous injection (0.1 ml at a rate of 0.4 ml/min). Mice were randomly assigned to receive cell lines or treatments. Endpoints for 3D-outgrowth and pulmonary tumor assays were determined prospectively. Growth was normalized to an initial reading taken 24 h post-plating (in vitro) or immediately after inoculation (in vivo).

## Quantitative real-time PCR

D2.OR cells were nonenzymatically isolated from 3D-culture using the Cultrex 3D-Culture Cell Harvesting Kit (Trevigen), and RNA was extracted using the RNeasy Mini Kit (QIAGEN). For patient-derived xenograft and tumor biopsy specimens, tissues were homogenized in TRIzol reagent (1 ml TRIzol/100 mg tissue), followed by RNA extraction and removal of DNA with DNase I treatment (Invitrogen). qRT-PCR was carried out as described (Gooding et al, 2017) using the primers listed in Table S2.

## Immunoprecipitation and immunoblotting

D2.OR cells were isolated from 3D-culture and homogenized on ice in RIPA lysis buffer (50 mM Tris-HCl, 150 mM NaCl, 6 mM sodium deoxycholate, 1.0% NP-40, 0.1% SDS, pH 7.4) supplemented with protease inhibitor cocktail (Sigma-Aldrich) and phosphatase inhibitors (10 mM sodium orthovanadate, 40 mM $\beta$-glycerophosphate, 20 mM NaF). Lysates were cleared by centrifugation and subjected to immunoblot analysis as described (Gooding et al, 2017).

## RNA stability analysis

Total RNA was isolated from mutagenized D2.OR cells and reverse-transcribed to generate SLX4IP antisense cDNA (asSLX4IP). PCR-amplified asSLX4IP (see Table S2) was phosphorylated using T4 polynucleotide kinase (10 U/reaction; New England Biolabs) and ligated into Pme I–digested pcDNA3.1(+) (1 U/reaction; New England Biolabs) that had been dephosphorylated using calf intestinal alkaline phosphatase (1 U/reaction; New England Biolabs). This construct was subsequently transfected into HEK293T cells using the TransIT-LT1 Transfection Reagent (Mirus Bio). Translation was arrested by treating cells with actinomycin D (10 $\mu$g/ml; Sigma-Aldrich) for the indicated times. RNA abundance was quantified using qRT-PCR and normalized at each time point to eukaryotic translation initiation factor 2 subunit 1 (eIF2$\alpha$).

## Immunofluorescence (IF) and FISH

IF/FISH experiments were carried out as described (Zeng et al, 2018) using antibodies against SLX4IP (Sigma-Aldrich), PML (Santa Cruz Biotechnology), or $\gamma$H2AX (Cell Signaling Technology) combined with a telomere leading strand probe [5'-(CCCTAA)$_3$-3'] conjugated to cyanine-5 (PNA Bio). Fluorescence detection was accomplished using a secondary antibody conjugated to Alexa Fluor 488 (Invitrogen). Images were captured using a Leica TCS SP8 STED confocal microscope (Light Microscopy Imaging Core, CWRU). The cells were classified as APB[+] according to (Fasching et al, 2007).

## RNA immunoprecipitation (RIP)

RIP was carried out as previously described (Peritz et al, 2006). Telomerase was immunoprecipitated from whole-cell lysate using anti-TERT IgM (Invitrogen). To generate IgM-binding beads, Protein A Sepharose beads were washed with 0.1% sodium azide and conjugated to IgG raised against murine IgM using dimethylpimelimidate (DMP; 20 mM). Conjugated beads were washed with borate buffer (100 mM $H_3BO_3$, 75 mM NaCl, 25 mM borax (Na$_2$B$_4$O$_7$), pH 9.0, and 3M NaCl), and crosslinking reactions were quenched using ethanolamine (200 mM, pH 8.0). Quantitation of protein-bound TERC was accomplished by qRT-PCR using the primers listed in Table S2.

## Telomerase activity assay

Endogenous telomerase activity was quantified from D2.OR cell extracts as described (Nandakumar et al, 2012). [32]P signal was detected using a Typhoon FLA 9500 (GE Healthcare Life Sciences) and quantitated using Imagequant TL. Activity calculations were performed according to published reports (Latrick & Cech, 2010).

## Telomere restriction fragment measurement

Telomere restriction fragment analysis was performed using the TeloTAGGG Telomere Length Assay (Roche) according to the manufacturer's instructions. A total of 4 $\mu$g DNA was digested overnight with RsaI and HinfI at 37°C and electrophoresed through a 0.8% agarose gel in 1× TBE (100 mM Tris base, 100 mM boric acid, 2 mM EDTA) at 65 V for 4 h. DNA was then capillary-transferred onto a Hybond-N[+] membrane (GE Healthcare) in 20× SSC (3M NaCl, 3M sodium citrate) for 3 d. The transferred DNA was fixed by UV cross-linking, and the membrane was hybridized to a digoxigenin (DIG)-labeled synthetic telomere probe [(GGGTTA)$_4$] overnight at 42°C. After hybridization, the membrane was washed with stringent buffer I (2× SSC, 0.1% SDS) at room temperature for 10 min and twice with stringent buffer II (0.2× SSC, 0.1% SDS) at 50°C for 15 min before incubation with an alkaline phosphatase–conjugated anti-DIG antibody. After substrate exposure, the membrane was imaged using the Odyssey Fc Dual-Mode Imaging System (LI-COR). Image quantification was performed using LI-COR Image Studio according to our previous studies (Hernandez-Sanchez et al, 2019).

## C-circle amplification assay

Amplification and quantitation of telomeric extrachromosomal circles (C-circles) in 3D-cultured cells were performed as previously described (Henson et al, 2009; Lau et al, 2013). Total cellular DNA quantitation was performed with the QuantiFluor ONE dsDNA System (Promega). C-circle quantitation was accomplished using the standard curve method, using U2OS DNA to generate the standard curve. The ribosomal protein 36B4 was used as a single-copy gene for normalization of linear chromosomal content. $\phi$29-deficient reactions were included for each sample.

### Senescence-associated β-galactosidase staining

SA-β-gal activity was quantified in 3D-cultured D2.OR cells using $C_{12}$FDG staining coupled with flow cytometry as described (Debacq-Chainiaux et al, 2009). Fluorescence was detected using an Attune NxT flow cytometer (Thermo Fisher Scientific), with the resulting SA-β-gal activity reported as FL1 median fluorescence intensity for each condition.

### Chromatin immunoprecipitation

ChIP was carried out according to our previous work (Gooding et al, 2017). Antibodies against proteins of interest were conjugated to Protein A/G Sepharose beads overnight at 4°C. 25 μg of DNA and 0.2 μg antibody/μg DNA were used for each immunoprecipitation. Measurement of DNA–protein interaction was accomplished by qRT-PCR.

### Survival analysis

Kaplan–Meier curves were generated using the breast cancer-specific KM Plotter online interface (http://kmplot.com/analysis/) (Gyorffy et al, 2010). Expression levels of each queried gene were subjected to quantile normalization, and patients were assigned to one of two groups based on individual expression level relative to the median expression in each sample. Subtype analyses were accomplished by restricting patient cohorts to ER⁻/PR⁻/HER2⁻ (i.e., triple-negative) or ER⁻/PR⁻/HER2⁺ cases. In all analyses, ER status was derived from gene expression data to maximize statistical power.

### Apoptosis assay

D2.OR, 4T1 series, or human cells were treated with varying concentrations of 5-FdU or AZ20 for 3 d. The cells were allowed to recover for 24 h in drug-free media before quantitation of caspase-3/7 activity using the Caspase-Glo 3/7 Assay (Promega) according to the manufacturer's instructions. Data are reported for each cell line as fold change in luminescence intensity relative to untreated cells.

### Colony formation assay

D2.OR cells (100 cells/well) were grown for 7 d, followed by treatment with 250 nM 5-FdU or 100 nM AZ20 for 3 d. After recovery in drug-free media, the cells were fixed in methanol:acetic acid (7:1 vol/vol) and stained with 0.5% crystal violet solution (BD Biosciences). Colonies were counted twice by two blinded individuals. Data are presented as mean number of colonies stained per well.

### Histology and immunohistochemistry

Lungs were fixed in 10% formalin before paraffin embedding and mounting of 5-μm sections on Superfrost Plus microscope slides (Thermo Fisher Scientific), which subsequently were (i) stained with hematoxylin and eosin (H&E), or (ii) subjected to γH2AX immunohistochemistry analysis with the Novolink Polymer Detection Systems (Leica Biosystems) according to the manufacturer's instructions.

### Gene expression profiling and gene set enrichment analysis (GSEA)

D2.OR cells were plated in 3D-culture and total RNA was isolated as described above. cDNA was labeled using the GeneChip WT Terminal Labeling Kit (Applied Biosystems) according to the manufacturer's instructions. Labeled cDNA was hybridized to GeneChip Mouse Gene 2.0 ST Arrays (three arrays per cell line; Affymetrix). Expression data were analyzed using the Transcriptome Analysis Console v. 4.0 (Affymetrix). Differentially expressed genes were called using a transcript abundance ratio (shSLX4IP:parental) ≥2 or ≤0.5 at a significance threshold $P < 0.05$. Microarray data can be found in the Gene Expression Omnibus under the accession number GSE125702. GSEA was carried out by querying significantly differentially expressed genes against the Molecular Signatures Database (MSigDB) collection C2 using GSEA software obtained from the Broad Institute. Gene expression and clinical data from The Cancer Genome Atlas were curated using cBioPortal (http://www.cbioportal.org/).

### Study approval

All animal studies were performed in accordance with the Institutional Animal Care and Use Committees for Case Western Reserve University. All studies involving human samples were approved by the Case Western Reserve University Institutional Review Board (IRB; UHCMC IRB Number: 01-13-43C). Informed consent was obtained from all patients in these studies, and all samples were de-identified before our acquisition.

### Statistical analysis

Where mean or median were used as measures of central tendency, statistical significance was determined using a two-sided Mann–Whitney $U$ test for single comparisons or Kruskal–Wallis test for multiple comparisons. RNA decay rate was estimated using a simple linear regression model of RNA abundance $R$ versus time $t$, and correlation was tested by applying a Fisher $Z$-transformation to the Spearman rank correlation coefficient (Spearman $\rho$) for $R(t)$. Nominal $p$- and FDR $q$-values for GSEA were calculated as described (Subramanian et al, 2005). For survival analysis, significance was determined using a log-rank (Mantel–Cox) test. For all experiments, $P < 0.05$ was considered statistically significant, with a Bonferroni correction applied post hoc for multiple comparisons. Unless otherwise noted, data are represented as mean ± SEM and are reflective of at least two independent experiments.

# Supplementary Information

# Acknowledgements

We thank all members of the Schiemann and Taylor laboratories for their critical input throughout this project, including preparation of the manuscript. Research support was provided in part by the National Institutes of Health (NIH) to WP Schiemann (CA177069 and CA236273), DJ Taylor (GM133841), and NJ Robinson (T32 GM007250 and F30 CA213892). Additional support was graciously provided by the Case Comprehensive Cancer Center's Research Innovation Fund, which is supported by the Case Council and Friends of the Case Comprehensive Cancer Center (WP Schiemann). We are grateful for the expertise and technical assistance provided by the Case Comprehensive Cancer Center Core Facilities (P30 CA43703), including the Gene Expression and Genotyping Facility, Imaging Research Core, Tissue Resources Core, and Cytometry and Imaging Microscopy Core. Microscopy experiments were performed in conjunction with the Case Light Microscopy Imaging Core, which is supported by an NIH Shared Instrumentation Grant (S10 OD016164).

## Author's Contributions

NJ Robinson: conceptualization, data curation, formal analysis, investigation, and methodology.
CD Morrison-Smith: formal analysis, investigation, and methodology.
AJ Gooding: formal analysis and investigation.
BJ Schiemann: formal analysis and investigation.
MW Jackson: methodology.
DJ Taylor: methodology.
WP Schiemann: conceptualization, formal analysis, investigation, and methodology.

## Conflict of Interest Statement

The authors declare that they have no conflict of interest.

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
