## [Reviewer comments · Life Science Alliance]

Life Science Alliance

SLX4IP and Telomere Dynamics Dictate Breast Cancer Metastasis and Therapeutic Responsiveness

William Schiemann, Nathaniel Robinson, Chevaun Morrison-Smith, Alex Gooding, Barbara Schiemann, Mark Jackson, and Derek Taylor

DOI: <https://doi.org/10.26508/lsa.201900427>

Corresponding author(s): William Schiemann, Case Western Reserve University

Review Timeline:

Submission Date:	2019-05-14
Editorial Decision:	2019-06-20
Revision Received:	2019-10-13
Editorial Decision:	2019-10-24
Revision Received:	2020-02-07
Editorial Decision:	2020-02-10
Revision Received:	2020-02-10
Accepted:	2020-02-11

Scientific Editor: Andrea Leibfried

Transaction Report:

June 20, 2019

Re: Life Science Alliance manuscript #LSA-2019-00427-T

Dr. William P Schiemann
Case Western Reserve University
General Medical sSciences (oncology)
2103 Cornell Rd
Cleveland, OH 44106

Dear Dr. Schiemann,

Thank you for submitting your manuscript entitled "SLX4IP and Telomere Dynamics Dictate Breast Cancer Metastasis and Therapeutic Responsiveness" to Life Science Alliance. The manuscript was assessed by expert reviewers, whose comments are appended to this letter.

As you will see, the reviewers appreciate your work, but note inconsistencies and the need to better support your conclusions. We would thus like to invite you to submit a revised version to us. Importantly, reviewer #2 raises very important points, which need to get satisfactorily addressed (except for point 3 - request for mechanistic insight - which is not mandatorily needed for publication here). We will need strong support from this reviewer on the revised version in order to move towards publication. Reviewer #1 requests reporting the outcome of injecting other clones - please do if data are available, otherwise please justify why this experiment was only done for three clones. All other points of this reviewer as well as the points raised by reviewer #3 should get addressed.

Thank you for this interesting contribution to Life Science Alliance. We are looking forward to receiving your revised manuscript.

Sincerely,

B. MANUSCRIPT ORGANIZATION AND FORMATTING:

Reviewer #1 (Comments to the Authors (Required)):

This article by Robinson et al and entitled "SLX4IP and Telomere Dynamics Dictate Breast Cancer Metastasis and Therapeutic Responsiveness" demonstrates that SLX4IP is a potential biomarker to predict breast tumor progression and metastasis.

The authors used validation based insertional mutagenesis to identify genes involved in tumor metastasis. The authors obtained 48 putative metastatic clones by using 3D cultures, but only injected 3 of them on mice (what about the other clones?). The VBIM 2-1 clone exhibited robust metastatic outgrowth and revealed SLX4IP as the VBIM strategy associated transcript. This was associated with a 50% decrease of the transcript expression and usage of shRNA strategy to decrease SLX4IP expression by using another strategy than VBIM demonstrated similar results in 3D culture and also observed a strong pulmonary tumor formation.

At the molecular level, the SLX4IP Knockdown is markedly associated to a transcriptional regulation of genes that negatively regulates metastasis (the list of the top regulated genes analysis by array need to be provided).

Functionally SLX4IP localize to telomeres and is associated to telomere maintenance. SLX4IP KD induces increased TERT expression and telomerase holoenzyme activity. Interestingly, SLX4IP inactivation impact on tumor metastasis requires the telomerase. Clinically, the authors clearly demonstrated that SLX4IP and TERT expression are inversely correlated in breast cancer with distinct pattern between TNBC and HER2 positive breast cancer. Moreover, this inverse expression pattern is always correlated to poor clinical outcomes for the patients.

Finally, the authors investigated about the potential of small molecules that target TMM to prevent tumor metastasis in the context of SLX4IP dysregulation. The use of AZ20 blocked the proliferation of D2.OR cells but not the SLX4IP KD counterpart. Conversely SLX4IP deficient cells were more sensitive to Floxuridine.

Overall the results of this paper are supporting that SLX4IP may serve a prime role in telomere maintenance and that alteration of its expression is playing a role in tumor metastasis. This is a good research article supported by strong evidences. The discovery presented herein may open new therapeutic strategy in the treatment of certain breast cancer subtypes

Reviewer #2 (Comments to the Authors (Required)):

In this manuscript, an insertional mutagenesis screen (VBIM) was used to identify SLX4IP as a mediator of metastatic recurrence. The authors then demonstrate that SLX4IP mediates metastasis by modulating telomere maintenance mechanism. Specifically, SLX4IP depletion suppressed ALT activity and coincided with activation of telomerase, with telomere maintenance mechanism preference impacting metastatic progression and patient survival. TMM-specific small molecule inhibitors were then used to further modulate telomere maintenance mechanism, although I am not convinced by the telomere maintenance mechanism-specificity of the small molecule inhibitors (and felt this wasn't fully demonstrated). The authors suggest that SLX4IP status (and telomere maintenance mechanism) can be used as a marker for breast cancer metastasis and therapeutic responsiveness.

1. The authors suggest that the D2.OR cell line is ALT to start with. This is a mouse cell line. Mouse cell lines have C-circles in all tissues (without ALT activity), as well as expressing telomerase. This is

likely to affect the findings. A more thorough characterisation of telomere maintenance mechanisms in mouse cells would be of use.

2. It is not appropriate to compare C-circle levels between human and mouse lines in Fig 2F.

3. I would like to see further mechanistic analysis of how SLX4IP deficiency causes the reduction in expression of genes that inhibit metastasis, and an increase in expression of TERT.

4. The data show an inverse correlation between SLX4IP and TERT in human TNBC compared to HER2+ breast cancer. Does this actually correlate with ALT/telomerase activity, ie do human triple-negative breast cancer cell lines typically use ALT (what is the frequency of ALT/telomerase)? From my understanding, the number of ALT breast cancer lines is very low. This would impact the therapeutic significance and is integral to the interpretation of the findings. The data currently don't address this, but rather look only at inverse expression patterns and patient outcomes.

5. I would like to see telomere length data to confirm telomere maintenance. Telomere maintenance mechanisms are inadequately characterised.

6. What is "relative TERT:TR holoenzyme" (Fig 2D)? How is this measured?

7. ALT cells are not specifically sensitive to ATR inhibitors.

8. Why is 5-FdU only incorporated into telomeres, and only by telomerase? I would like to see evidence of this. Presumably it can be incorporated elsewhere and during ALT telomere lengthening.

9. The ALT and telomerase-specific inhibitors are inadequately characterised.

Reviewer #3 (Comments to the Authors (Required)):

This study highlights that SLX4-Interacting protein (SLX4IP) correlates with telomere homeostatic pathways, involves in metastatic outgrowth of DTCs in some breast cancer subtypes, and potentially can be used as a predictive marker of cancer metastasis and survival. Furthermore, by pharmacologic targeting of molecular indicators of SLXIP/TERT pathways, they show some therapeutic benefits in their in vivo models. Although, the findings of this study are important for understanding the breast cancer metastasis and therapy, a revised version of manuscript is needed to be seen before acceptance.

Major Comments:

1- In Fig.1 the knock-down expression experiments represented by RNA level measurement and that would be more convincing to show protein expression by western blot or other relevant techniques. Authors may explain limitations if any.

2- Figure 1.H "gene set enrichment analysis" is non-informative (e.g. it lacks the name of the genes etc. ...)

3- For easier understanding, SLX4IP rescue experiment (S.2 C) should come with Fig. 2G.

4- Page 9 of manuscript referred ' supplementary Fig. S4' does not match the text all through- out the page.

5- Rescue studies in their knock-out human breast cancer lines (fig. 5) would be appealing.

6- No consistency in their in vivo injection protocols. For example tail-vein injection for screening VBIM metastatic mutants (Materials and Methods) vs Intravenous inoculation (page 12) for drug treatment. Are they different or just different terminology for the same injection protocol?

7- The findings and conclusions support the potential role of SLX4IP/TERT axis in tumor outgrowth of only some subtypes of studied human TNBC. However the title is so broad and doesn't fully fit with their findings.

RE: Revisions for Manuscript # LSA-2019-00427-T

Dear Dr. Leibfried,

My colleagues and I would like to thank you and the Reviewers for their comments on our manuscript titled "*SLX4IP and Telomere Dynamics Dictate Breast Cancer Metastasis and Therapeutic Responsiveness*" (LSA-2019-00427-T). Indeed, we were pleased to learn that LSA desired a revised version of the manuscript, and that the Reviewers believed our study "is a good research article supported by strong evidences." Moreover, it was noted by the Reviewers that the findings of our "study are important for understanding breast cancer metastasis," and that our discovery "may open new therapeutic strategy in the treatment of certain breast cancer subtypes." Despite the overall enthusiasm for this study, the Reviewers noted several minor-to-moderate concerns that precluded acceptance of our manuscript to LSA. As such, I would like to take a moment of your time to address the issues raised by the Reviewers, and to describe on a point-by-point basis how we responded to them.

Reviewer 1:

Point 1: *The authors used validation based insertional mutagenesis to identify genes involved in tumor metastasis. The authors obtained 48 putative metastatic clones by using 3D cultures, but only injected 3 of them on mice (what about the other clones?).*

Our Response: The Reviewer's point is well-taken and we greatly appreciate the curiosity directed towards the identity and nature of these additional clones in regulating metastatic recurrence. Unfortunately, we have not vigorously pursued the identity and mechanistic features of these clones, focusing all of our efforts on SLX4IP and trying to discover how this scaffold protein elicits metastatic relapse. Having said that, we have identified 1 additional genomic locus targeted by VBIM and have initiated *in vitro* characterization/validation analyses related to 3D-outgrowth. Clearly, a thorough characterization of this clone and others lies beyond the scope of the current study, and we look forward to pursuing and reporting these clones in the future.

Point 2: *At the molecular level, the SLX4IP Knockdown is markedly associated to a transcriptional regulation of genes that negatively regulates metastasis (the list of the top regulated genes analysis by array need to be provided).*

Our Response: We thank the Reviewer for bringing this omission to our attention. We now provide the requested gene list in Fig. S2.

Reviewer 2:

Point 1: *The authors suggest that the D2.OR cell line is ALT to start with. This is a mouse cell line. Mouse cell lines have C-circles in all tissues (without ALT activity), as well as expressing telomerase. This is likely to affect the findings. A more thorough characterisation of telomere maintenance mechanisms in mouse cells would be of use. In Fig 2d (chart on right), control should be shown on the left of the paired conditions.*

Our Response: The Reviewer's concerns regarding the differences between mouse and human telomere dynamics are well-founded. We too had these concerns and sought to allay them by employing multiple measures and metrics for each TMM. For instance, we measured ALT not only by C-circle abundance (Fig. 3C), but also by (i) quantifying the presence of ALT-associated PML bodies (Fig. 3A & 3B); and (ii) monitoring the expression levels (Fig. 3D) and epigenetic states (Fig. 3E) of ATRX and Daxx. With respect to telomerase, we assessed the expression (Fig. 2B & 2E), holoenzyme abundance (Fig. 2C), and activity (Fig. 2D) of TERT. Additionally, while mouse lines do indeed show evidence of both TMMs simultaneously, our findings clearly demonstrate a shift from primarily ALT-like to TERT-like phenotypes in cells rendered SLX4IP-deficient. Equally important, our findings indicate that this dramatic shift overrides any residual activity of a secondary TMM, and as such, portends to critical differences in the tumorigenicity and therapeutic response of these breast cancer cells.

Point 2: *It is not appropriate to compare C-circle levels between human and mouse lines in Fig 2F.*

Our Response: We understand the Reviewer's concern, which is in many respects an extension of Point 1. It is important to note that whether mouse cells contain C-circles is in many respects irrelevant to our study. The critical point is that inactivation of SLX4IP elicits a strong and repeatable reduction in C-circle content indicative of suppression of ALT not only in mouse cell lines, but also a variety of human breast cancer cells. As discussed above, we monitored alterations in TMMs using several complementary strategies across our mouse and human cell models. Finally, the C-circle comparison shown in Fig. 3C is provided to demonstrate our proficiency in accurately detecting C-circles in cells known to harbor ALT (e.g., U2OS) versus those driven by TERT (e.g., MCF-7). These positive and negative controls lend credence to our ability to measure alterations in C-circle content that transpire in response to targeted inactivation of SLX4IP expression. Taken together, these strategies and their findings support the major conclusions of the study.

Point 3: *I would like to see further mechanistic analysis of how SLX4IP deficiency causes the reduction in expression of genes that inhibit metastasis, and an increase in expression of TERT.*

Our Response: The Reviewer's point is well-taken and we too remain intrigued in elucidating how SLX4IP functions in mediating ALT, as well as how loss of SLX4IP promotes a shift to TERT. At present, we have identified several signaling pathways that lie upstream of SLX4IP in ALT⁺ cells, as well as those that become engaged upon the loss of SLX4IP as cells activate TERT expression and activity. Unfortunately, the precise mechanistic triggers that shunt cells towards one TMM versus another remains mysterious. We believe these mechanistic analyses need to be worked out in depth, and as such, a thorough characterization of these events is clearly beyond the scope of this study.

Point 4: *The data show an inverse correlation between SLX4IP and TERT in human TNBC compared to HER2+ breast cancer. Does this actually correlate with ALT/telomerase activity, ie do human triple-negative breast cancer cell lines typically use ALT (what is the frequency of ALT/telomerase)? From my understanding, the number of ALT breast cancer lines is very low. This would impact the therapeutic significance and is integral to the interpretation of the findings. The data currently don't address this, but rather look only at inverse expression patterns and patient outcomes.*

Our Response: We thank the Reviewer for raising this interesting and complex concern. Indeed, the relative frequencies of each TMM within and across genetically distinct breast cancer subtypes remains to

be investigated in a comprehensive and rigorous manner. However, evidence does exist that suggests that a proportion of HER2⁺ breast cancers do in fact utilize ALT (see Subhawaong et al, 2009). At present, it is difficult to assess the predominant TMM used by cells and tissues based on publicly available gene expression profiles because established “hallmark” gene signatures uniquely associated with ALT remain to be fully elucidated. Based on current evidence, it is perhaps best and most accurate to describe the frequency of ALT in breast cancer cell lines as “unknown” as opposed to low, as a comprehensive survey of TMM identity that employs APB quantification, etc. has not been conducted. In considering these limitations, we nevertheless believe our findings assert SLX4IP as a candidate marker of ALT in breast cancer based on our findings of APB staining and relation to TERT expression. While a more comprehensive ALT signature could potentially be derived from gene expression data, such an endeavor clearly falls outside the scope of this study.

Point 5: *I would like to see telomere length data to confirm telomere maintenance. Telomere maintenance mechanisms are inadequately characterised.*

Our Response: We appreciate the Reviewer’s concern and concur that this aspect needed to be bolstered in our study. To do so, we undertook telomere restriction fragment (TRF) analysis on our D2.OR derivatives. These new and important findings are included in our **revised Fig. 4D**. It is important to note that these data reveal the presence of longer telomeres in parental D2.OR cells as compared to their SLX4IP-depleted counterparts, findings consistent with a switch in TMM from ALT to TERT following SLX4IP knockdown. Moreover, CRISPR/Cas9-mediated inactivation of TERT in SLX4IP-depleted cells brought about substantial telomere shortening as would be expected in cells lacking both ALT and TERT. It should also be noted that the calculated telomere lengths are shorter than would normally be expected for murine cell lines. However, several recent studies described mouse telomeres as being long were often conducted using primary mouse cells or tissues (Zijlmans et al, 1997). In contrast, several studies found that established murine cell lines possess telomeres that are similar in length both to human telomeres, and to those measured in our D2.OR derivatives (Marie-Egyptienne et al, 2008; McIlrath et al, 2001; Sachsinger et al, 2001). Collectively, these findings, together with those included in Figs. 2 and 3, provide robust characterization of the TMM identities harbored by these cells.

Marie-Egyptienne DT, ME Brault, S Zhu, and C Autexier (2008) Telomerase inhibition in a mouse cell line with long telomeres leads to rapid telomerase reactivation. *Exp Cell Res* 314:668-675.

McIlrath J, SD Bouffler, E Samper, A Cuthbert, A Wojcik, I Szumiel, PE Bryant, AC Riches, A Thompson, MA Blasco et al. (2001) Telomere length abnormalities in mammalian radiosensitive cells. *Cancer Res* 61:912-915.

Sachsinger J, E Gonzalez-Suarez, E Samper, R Heicappell, M Muller, and MA Blasco (2001) Telomerase inhibition in RenCa, a murine tumor cell line with short telomeres, by overexpression of a dominant negative mTERT mutant, reveals fundamental differences in telomerase regulation between human and murine cells. *Cancer Res* 61:5580-5586.

Zijlmans JM, UM Martens, SS Poon, AK Raap, HJ Tanke, RK Ward, and PM Lansdorp (1997) Telomeres in the mouse have large inter-chromosomal variations in the number of T2AG3 repeats. *Proc Natl Acad Sci U S A* 94:7423-7428.

Point 6: *What is “relative TERT:TR holoenzyme” (Fig 2D)? How is this measured?*

Our Response: We thank the Reviewer for bringing this clarification to our attention and apologize for not providing a better description of this term. In correcting this oversight, we have modified the corresponding text in the “Results” section to better define this term by stating the this measure reflects **“the core telomerase holoenzyme (i.e., TERT plus the telomerase RNA component TR).”** We have also modified the legend for Fig. 2C to include additional experimental detail for measuring **“the mature telomerase core holoenzyme following RNA immunoprecipitation of TERT-bound TR,”** which is further elaborated in the corresponding “Materials and Methods” section.

Point 7: *ALT cells are not specifically sensitive to ATR inhibitors.*

Our Response: The Reviewer is correct in noting the controversy regarding the sensitivity of ALT⁺ cells to ATR inhibitors. We too are aware of this ongoing debate and have attempted to steer our way

through this dilemma using additional experimentation and textual clarifications. Experimentally, we utilized multiple ATR inhibitors (e.g., AZ20 and VE-821) and observed both compounds to be capable of repressing the growth of breast cancer organoids (Figs. 6A & 6B), and of reducing APB abundance (Fig. 6E). Importantly, the ability of these compounds to reduce 3D-outgrowth and APB abundance was solely limited to ALT-positive D2.OR cells. In a separate line of research, we undertook an ATR-independent strategy that is predicted to preferentially target ALT cells – namely, the administration of the small molecule BLM inhibitor (ML216; Fig. 6B). Importantly, targeting of BLM suppressed the outgrowth of ALT⁺ D2.OR cells in a manner reminiscent of that elicited by ATR inhibition. Finally, we have updated our “Results” section to explicitly note the conflicting studies regarding ATR inhibition in cells reliant upon ALT cells. Collectively, we believe these results support a connection between SLX4IP and TMM identity and provide a basis for employing TMM-targeting therapeutic strategies.

Point 8: *Why is 5-FdU only incorporated into telomeres, and only by telomerase? I would like to see evidence of this. Presumably it can be incorporated elsewhere and during ALT telomere lengthening.*

Our Response: We appreciate the Reviewer’s concern. However, Dr. Taylor, who is a coauthor on this study, published this information last year in Cell Reports (Cell Rep. 2018 Jun 5;23(10):3031-3041. DOI: 10.1016/j.celrep.2018.05.020; PMID: 29874588). Importantly, we significantly extend these findings by demonstrating the effectiveness of 5-FdU in preclinical mouse models of TNBC metastasis, which contrasts sharply from the ineffectiveness of 5-FU to impact TNBC metastasis.

Point 9: *The ALT and telomerase-specific inhibitors are inadequately characterised.*

Our Response: With all due respect, we are unclear as to the Reviewer’s objections on this point. Indeed, the specificity and kinetic properties of the telomerase inhibitor, BIBR2532 (Fig. S6A) have been established previously. Our other telomerase-targeting drug, 5-FdU, does not function as a direct inhibitor of TERT activity (see Major Point 8). Indeed, the activity of telomerase is required for this compound to exert its cytotoxic effects (Fig. 6G-H and S6A). For other considerations regarding the specificity of 5-FdU, please see Point 8. With respect to ALT inhibitors, we showed that the AZ20 concentrations used herein inhibit ATR activity in response to replication fork stalling (Fig. S4A), while failing to inhibit other PI3K-related kinases (Fig. S4B). As such, we remain unsure how to further pursue this concern without additional direction/clarification.

Reviewer 3:

Point 1: *In Fig.1 the knock- down expression experiments represented by RNA level measurement and that would be more convincing to show protein expression by western blot or other relevant techniques. Authors may explain limitations if any.*

Our Response: The Reviewer’s point is well-taken and we now provide evidence of SLX4IP mRNA and protein abundance in our revised Fig. 1A.

Point 2: *Figure 1.H "gene set enrichment analysis" is non-informative (e.g. it lacks the name of the genes etc. ...).*

Our Response: We thank the Reviewer for bringing this point to our attention. Please see Reviewer 1, Point 2 for how we addressed this concern.

Point 3: *For easier understanding, SLX4IP rescue experiment (S.2 C) should come with Fig. 2G.*

Our Response: We thank the Reviewer for this insightful suggestion. Indeed, we have now streamlined the presentation of these data according to the Reviewer’s recommendation. Panels previously labeled as Fig. 2G and Fig. S2C have been combined and are now included in Fig. 3A.

Point 4: *Page 9 of manuscript referred 'supplementary Fig. S4' does not match the text all throughout the page.*

Our Response: We thank the Reviewer for pointing out this inconsistency. All Figures have now been properly matched with their supporting Supplementary Figures throughout the manuscript.

Point 5: *Rescue studies in their knock-out human breast cancer lines (fig. 5) would be appealing.*

Our Response: The Reviewer's point is well-taken and very important. To address this concern, we now show the impact of reintroducing SLX4IP expression in human and murine cells rendered deficient in this scaffolding protein. Importantly, re-expression of SLX4IP reinstated the ALT characteristics, growth properties, and sensitivities to ATR inhibitors (See Figs. 1 & 2).

Point 6: *No consistency in their in vivo injection protocols. For example tail-vein injection for screening VBIM metastatic mutants (Materials and Methods) vs Intravenous inoculation (page 12) for drug treatment. Are they different or just different terminology for the same injection protocol?*

Our Response: We appreciate the Reviewer for bringing this ambiguity and oversight to our attention. The VBIM subsection of the "Materials and Methods" has been modified to include the word "*intravenous*" when describing the *in vivo* arm of our genetic screen in order to highlight the fact that all injections were performed in the same manner throughout this study.

My colleagues and I again would like to thank you and the Reviewers for their comments and criticisms, which were instrumental in strengthening the importance and significance of our manuscript. We truly appreciate your efforts on behalf of this manuscript, and we look forward to hearing from you and learning of the study's acceptance to *Life Science Alliance*.

October 24, 2019

Re: Life Science Alliance manuscript #LSA-2019-00427-TR

Dr. William P Schiemann
Case Western Reserve University
General Medical sSciences (oncology)
2103 Cornell Rd
Cleveland, OH 44106

Dear Dr. Schiemann,

Thank you for submitting your manuscript entitled "SLX4IP and Telomere Dynamics Dictate Breast Cancer Metastasis and Therapeutic Responsiveness" to Life Science Alliance. The manuscript was assessed by reviewer #2 again, whose comments are appended to this letter.

As you know from our previous email exchange, we would need strong support from reviewer #2 on the revised version of your work in order to move forward here. Unfortunately, and as you will see below, reviewer #2 is not entirely happy with the revision and thinks that her/his previous points 1, 2, 4, 5 and 8 were not well addressed.

We think that the remaining concern regarding points 1, 2, 4, and 8 can get addressed by re-wording of your rebuttal and by acknowledging that 5-FdU may get incorporated elsewhere than at telomeres but that you see telomerase-specific effects. The remaining concern regarding point 5, however, needs to get addressed with better data. We usually allow only a single round of experimental revision, but if you think you'll be able to provide better quality telomere length data with more of the gel shown and at an exposure that makes the relative telomere lengths clear, we'd be happy to consider your work further here. Please discuss the short telomeres in the text as well and:

- Please provide the supplementary figures as separate files, their legends should go into the main manuscript file
- Please provide the supplementary tables in word docx or excel format
- Please add callouts in the text to Fig S5E-F (see page 11)
- Please add next to the p-values in the figure legends which statistical test was used

We are looking forward to receiving your revised manuscript.

Sincerely,

Andrea Leibfried, PhD
Executive Editor

Life Science Alliance
Meyerhofstr. 1
69117 Heidelberg, Germany
t +49 6221 8891 502
e a.leibfried@life-science-alliance.org
www.life-science-alliance.org

B. MANUSCRIPT ORGANIZATION AND FORMATTING:

Reviewer #2 (Comments to the Authors (Required)):

The authors have done a reasonable job in addressing the concerns of the reviewers. I have some continuing concerns. First, I disagree with the comment that "whether mouse cells contain C-circles is in many respects irrelevant to our study", when the following sentence reads "The critical point is that inactivation of SLX4IP elicits a strong and repeatable reduction in C-circle content indicative of suppression of ALT not only in mouse cell lines..." (Point 2). Point 4 could have been addressed by modifying the text to mention TMM prevalence, and I feel that this should be done. Point 5: the TRF

now included in Fig 4D is very poor quality. Why is there a decrease in telomere length in the Scram control? And these telomere lengths are very short. This is mentioned (with references) in the rebuttal, but not in the text. I don't think the TRF is in any way indicative of a change in TMM - in fact there doesn't seem to be a TMM following SLX4IP depletion. Point 8: The Cell Reports paper did not address whether ALT can result in telomeric incorporation of 5-FdU. I would imagine that misincorporation of 5-FdU during break-induced telomere synthesis is entirely possible. This should be considered.

Dr. Andrea Leibfried

Executive Editor, Life Science Alliance

RE: Revisions for Manuscript # LSA-2019-00427-T

Dear Dr. Leibfried,

My colleagues and I would like to thank you and the Reviewers for their comments on our manuscript titled “*SLX4IP and Telomere Dynamics Dictate Breast Cancer Metastasis and Therapeutic Responsiveness*” (LSA-2019-00427-T). Indeed, we were pleased to learn that LSA desired a revised version of the manuscript, and that the Reviewers 1 & 3 raised no additional issues or concerns regarding our study. Although Reviewer 2 acknowledged that we did “a reasonable job in addressing the concerns of the reviewers,” this expert nevertheless still has “some continuing concerns” regarding our study and its acceptability to LSA. As such, I would like to take a moment of your time to address the issues raised by the Reviewer 2 and the Editorial Board, and to describe on a point-by-point basis how we responded to them.

Reviewer 2 & Editorial Concerns:

We greatly appreciate the provided insights into how to best address these concerns, which are discussed below.

Point 2: *First, I disagree with the comment that “whether mouse cells contain C-circles is in many respects irrelevant to our study”, when the following sentence reads “The critical point is that inactivation of SLX4IP elicits a strong and repeatable reduction in C-circle content indicative of suppression of ALT not only in mouse cell lines...” (Point 2).*

Our Response: We concur with Reviewer 2 that it is inappropriate to directly compare C-circle content between human and murine cell lines, as shown in **Fig. 3C**. In our **revised Fig. 3C**, we now present left (human) and right (murine) data panels, wherein human C-circle content is normalized to that present in U2OS cells (*left panel*), and conversely, murine C-circle content is normalized to that present in parental D2.OR cells (*right panel*). We believe that these species-specific comparisons are more representative of the measured differences in C-circle context that we detect in response to SLX4IP inactivation.

Point 4: *Point 4 could have been addressed by modifying the text to mention TMM prevalence, and I feel that this should be done.*

Our Response: We thank Reviewer 2 for highlighting this important point, which we addressed on **page 16** of our revised manuscript by stating (revised text is in **red**):

TMM identity is classically viewed as a stable property of immortalized cells. More accurately, immortalization achieved during neoplastic transformation is generally presumed to be carried out by telomerase, with ALT serving as a reserve mechanism that becomes operational when telomerase function is disrupted. More recent examinations have uncovered pathologic evidence of ALT in ~15% of cancers, most frequently in tumors of mesenchymal origin, such as osteosarcomas and gliomas. Notably, however, ALT has also been detected

in cancers of the bladder, cervix, endometrium, esophagus, kidney, liver, and lung, and in non-glioma CNS tumors (Heaphy et al, 2011b). In line with these observations, evidence of ALT is observed in a subset (~15%) of HER2-enriched breast cancer patients who presented with lymph node metastases at the time of initial diagnosis and ultimately succumbed to highly aggressive disease (Subhawong et al, 2009). In contrast, no evidence of ALT was found in TNBC patients, suggesting that the telomeres in these tumors were maintained by telomerase. These findings are consistent with our assertion that a SLX4IP^{High}/TERT^{Low} gene expression profile is indicative of ALT and associated with poor outcomes specifically in HER2-enriched breast cancer patients (Fig 5). Our investigation also asserts the possible existence of an innate plasticity in TMM selection. Moreover, the relationship between SLX4IP and TMM acquisition likely represents only one facet of a complex regulatory network that receives inputs from a multitude of cell-intrinsic and microenvironmental cues, events likely to be honed by signaling inputs derived from Wnt/ β -catenin, NF- κ B (Yin et al, 2000), and c-Myc (Wu et al, 1999). As such, future studies need to explore (i) the significance of SLX4IP and ALT as a driver of tumor progression, (ii) the plasticity inherent in the establishment and preservation of TMM identity, and (iii) the regulatory landscape of SLX4IP and its connections to the signaling pathways listed above.

Point 5: *The TRF now included in Fig 4D is very poor quality. Why is there a decrease in telomere length in the Scram control?*

Our Response: Reviewer 2 is justified in criticizing the poor quality of our original TRF blot. To address this issue, we worked extremely hard to improve several technical aspects of the analyses, including altering (i) agarose percentage of gels; (ii) increased wash stringencies; and (iii) increased gel transfer times. Additionally, we determined that these analyses work best on cells that are actively growing over several passages in culture as compared to those used immediately upon thawing. In our revised manuscript, we now present 2 examples of TRF analyses. Example 1 is provided in **Fig. 2H** and shows that loss of SLX4IP expression initially results in telomere attrition, followed by significant extension that coincides with TERT expression. Example 2 is provided in **Fig. 4D** and shows that inactivation of both SLX4IP and TERT only results in telomere attrition that culminates in cellular senescence. Overall, these analyses greatly enhance the overall significance and impact our findings, and as such, we wholeheartedly thank the Reviewer for remaining steadfast that we improve the fidelity of these analyses.

Point 8: *The Cell Reports paper did not address whether ALT can result in telomeric incorporation of 5-FdU. I would imagine that misincorporation of 5-FdU during break-induced telomere synthesis is entirely possible. This should be considered.*

Our Response: We again thank Reviewer 2 for highlighting this important point, which we addressed on **page 18** of our revised manuscript by stating (revised text is in red):

Nevertheless, targeting TMMs remains an appealing therapeutic strategy in need of novel approaches, such as utilizing nucleoside analogs that act as substrates for telomerase in a manner that is completely distinct from telomerase inhibition. Indeed, we found that 5-FdU co-opts telomerase activity to initiate cell death in telomerase-positive breast cancer cells and eradicate telomerase-driven metastatic disease (Fig 6-8). The beneficial effects of 5-FdU on metastatic breast cancer cells appear to be dependent upon telomerase, as evidenced by the loss of therapeutic efficacy following genetic ablation (Fig 6G and H) or pharmacologic inhibition (Fig S6A) of TERT. However, it remains possible that the 5-FdU is also misincorporated during break-induced telomere synthesis, an event that would conceivably produce deleterious effects on ALT cells. Although by no means definitive, this concern is partially mitigated by two important observations. First, 5-FdU does not inhibit the growth of U2OS cells (Fig S6A). Second, 5-FdU is incorporated into telomeres by telomerase when administered at doses below those at which it can be utilized by other DNA polymerases (Zeng et al, 2018). Thus, while future studies clearly need to examine the fate of 5-FdU in human breast cancers, our observations lend support to the notion that low-dose 5-FdU possesses high selectivity for telomerase and induces preferential

cytotoxicity in telomerase-driven cancers. Interestingly, our work also reveals a pharmacodynamic divergence between 5-FdU and 5-FU (Fig 6F, 8B and 8C), thereby shedding new mechanistic light upon 5-FdU and its potential clinical repurposing toward novel targets such as TMMs. Our study further suggests that ALT may serve as an adaptive mechanism that is preferentially activated by dormant DTCs. Importantly, this may provide a unique targeting strategy for abrogating recurrent disease that can be accomplished through combinatorial targeting of multiple pathways (e.g., anti-ATR or anti-BLM in combination with standard-of-care or anti-telomerase agents).

Additional Editorial Points:

Editorial Point 1: *Please provide the supplementary figures as separate files, their legends should go into the main manuscript file.*

Our Response: We have made the requested modifications to the Supplementary Figures and their legends.

Editorial Point 2: *Please provide the supplementary tables in word docx or excel format.*

Our Response: We have made the requested modifications to the Supplementary Tables.

Editorial Point 3: *Please add callouts in the text to Fig S5E-F (see page 11).*

Our Response: We apologize for this oversight and now include the aforementioned callouts.

Editorial Point 4: *Please add next to the p-values in the figure legends which statistical test was used.*

Our Response: We have made the requested modifications to all figure legends.

My colleagues and I again would like to thank you and the Reviewers for their comments and criticisms, which were instrumental in strengthening the importance and significance of our manuscript. We truly appreciate your efforts on behalf of this manuscript, and we look forward to hearing from you and learning of the study's acceptance to *Life Science Alliance*.

February 10, 2020

RE: Life Science Alliance Manuscript #LSA-2019-00427-TRR

Dr. William P Schiemann
Case Western Reserve University
General Medical sSciences (oncology)
2103 Cornell Rd
Cleveland, OH 44106

Dear Dr. Schiemann,

Thank you for submitting your revised manuscript entitled "SLX4IP and Telomere Dynamics Dictate Breast Cancer Metastasis and Therapeutic Responsiveness". I appreciate the introduced changes and the improved TRF assays provided, and we would thus be happy to publish your paper in Life Science Alliance. Please login one more time to fill in our electronic license to publish form (you need to move all files to the next manuscript version prior to do so, please - it's a single-click process).

A. FINAL FILES:

B. MANUSCRIPT ORGANIZATION AND FORMATTING:

Sincerely,

February 11, 2020

RE: Life Science Alliance Manuscript #LSA-2019-00427-TRRR

Dr. William P Schiemann
Case Western Reserve University
General Medical sSciences (oncology)
2103 Cornell Rd
Cleveland, OH 44106

Dear Dr. Schiemann,

Thank you for submitting your Research Article entitled "SLX4IP and Telomere Dynamics Dictate Breast Cancer Metastasis and Therapeutic Responsiveness". It is a pleasure to let you know that your manuscript is now accepted for publication in Life Science Alliance. Congratulations on this interesting work.

DISTRIBUTION OF MATERIALS:

Again, congratulations on a very nice paper. I hope you found the review process to be constructive and are pleased with how the manuscript was handled editorially. We look forward to future exciting submissions from your lab.

Sincerely,
